# Mosquito community composition shapes virus prevalence patterns along anthropogenic disturbance gradients

Kyra Hermanns[1], Marco Marklewitz[1], Florian Zirkel[2†], Anne Kopp[1], Stephanie Kramer-Schadt[3,4], Sandra Junglen[1*]

[1]Institute of Virology, Charité - Universitätsmedizin Berlin, corporate member of Free University Berlin, Humboldt-Universtiy Berlin, and Berlin Institute of Health, Berlin, Germany; [2]Institute of Virology, University of Bonn Medical Centre, Berlin, Germany; [3]Department of Ecological Dynamics, Leibniz Institute for Zoo and Wildlife Research, Berlin, Germany; [4]Institute of Ecology, Technische Universität Berlin, Berlin, Germany

**\*For correspondence:**
sandra.junglen@charite.de

**Present address:** [†]Biotest AG, Dreieich, Germany

**Competing interest:** The authors declare that no competing interests exist.

**Abstract** Previously unknown pathogens often emerge from primary ecosystems, but there is little knowledge on the mechanisms of emergence. Most studies analyzing the influence of land-use change on pathogen emergence focus on a single host–pathogen system and often observe contradictory effects. Here, we studied virus diversity and prevalence patterns in natural and disturbed ecosystems using a multi-host and multi-taxa approach. Mosquitoes sampled along a disturbance gradient in Côte d'Ivoire were tested by generic RT-PCR assays established for all major arbovirus and insect-specific virus taxa including novel viruses previously discovered in these samples based on cell culture isolates enabling an unbiased and comprehensive approach. The taxonomic composition of detected viruses was characterized and viral infection rates according to habitat and host were analyzed. We detected 331 viral sequences pertaining to 34 novel and 15 previously identified viruses of the families *Flavi-*, *Rhabdo-*, *Reo-*, *Toga-*, *Mesoni-* and *Iflaviridae* and the order *Bunyavirales*. Highest host and virus diversity was observed in pristine and intermediately disturbed habitats. The majority of the 49 viruses was detected with low prevalence. However, nine viruses were found frequently across different habitats of which five viruses increased in prevalence towards disturbed habitats, in congruence with the dilution effect hypothesis. These viruses were mainly associated with one specific mosquito species (*Culex nebulosus*), which increased in relative abundance from pristine (3%) to disturbed habitats (38%). Interestingly, the observed increased prevalence of these five viruses in disturbed habitats was not caused by higher host infection rates but by increased host abundance, an effect tentatively named abundance effect. Our data show that host species composition is critical for virus abundance. Environmental changes that lead to an uneven host community composition and to more individuals of a single species are a key driver of virus emergence.

## Editor's evaluation

This paper explores the drivers of viral and host composition in natural and disturbed ecosystems. The authors make an important contribution to knowledge on the diversity of mosquito-specific viruses, describing the genetic diversity of RNA viruses from the family *Culicidae*. The paper will be of interest to scientists in the fields of virology, entomology, ecology and epidemiology. The data are of high quality and have been rigorously assessed.

## Introduction

A major challenge in understanding the emergence of infectious disease is to identify the driving factors responsible for changes in infectious disease dynamics. New infectious diseases mostly emerge in tropical regions that have undergone strong ecological and economic change (*Swei et al., 2020*). Tropical rainforests are terrestrial ecosystems with a high biodiversity, constituting a habitat with a diverse range of hosts and pathogens. This high host richness likely corresponds to a high pathogen richness as each host is likely to carry its own specific pathogens (*Dunn et al., 2010*). Pristine rainforests are subject to large scale anthropogenic land use transformation leading to increased contact among humans, wildlife, and pathogens (*Keesing et al., 2010*; *Olival et al., 2017*). Disturbed habitats often show a drastic decline in biodiversity or turnover of host species community composition, which is accompanied by an increase of species that are resilient to disturbance, so-called generalist species. How changes in community composition influence pathogen prevalence in target hosts is still unclear and a matter of scientific debate (*Randolph and Dobson, 2012*).

In this context, the dilution effect hypothesis postulates that a high diversity of different host species dilutes the prevalence of a specific pathogen in intact ecosystems as the density of competent hosts (hosts that contribute to pathogen transmission and maintenance) is diluted by the presence of non-competent hosts. Biodiversity loss increases the infection prevalence of this specific pathogen in the disturbance-resilient, competent host species (*Ostfeld, 2017*). Prerequisites for the occurrence of a dilution effect are the presence of species that differ in their susceptibility for a particular pathogen and a lower risk of extinction of competent hosts under habitat disturbance. This may then lead to a higher relative frequency of competent host species in modified habitats (*Halliday et al., 2017*; *Joseph et al., 2013*; *Ostfeld and Keesing, 2012*). Hence, in the case of zoonotic pathogens, the disease risk for humans is increased (*Ostfeld, 2017*; *Gibb et al., 2020*). Studies confirming the dilution effect hypothesis focused on pathogens known to cause outbreaks in humans and livestock (e.g. West Nile virus [WNV], Sin Nombre virus, and *Borrelia* spp.) and use generalist species as hosts (*Allan et al., 2009*; *Clay et al., 2009*; *Ostfeld and Keesing, 2000*; *Khalil et al., 2016*). However, it seems that this effect cannot be generalized and that diverse mechanisms regulate infectious disease transmission dynamics in a host-, pathogen-, and situation-dependent manner (*Guo et al., 2019*; *Johnson et al., 2015b*). Some studies observed a contrary effect for WNV, Usutu virus, and *Borellia* spp., where the infection rate or density of infected hosts increased with biodiversity, referred to as amplification effect (*Levine et al., 2017*; *Ruyts et al., 2016*; *Roiz et al., 2019*).

These conflicting results indicate that biodiversity loss has complex impacts on disease risk and effects can be heterogeneous and scale-dependent (*Johnson et al., 2015b*; *Keesing et al., 2006*; *McLeish et al., 2017*; *Salkeld et al., 2013*). Partial views on single host species or single pathogens cannot reveal general mechanisms. Existing studies either focused only on pathogen discovery aside from an ecological context or on one specific pathogen in a specific host across different habitats (*Clay et al., 2009*; *Ostfeld and Keesing, 2000*; *Levine et al., 2017*; *Shi et al., 2016*). Studies with a wider scope analyzing community composition of entire host groups and assessing the genetic diversity of their pathogens in undisturbed and disturbed habitats are lacking and would be paramount for a comprehensive understanding of biodiversity-infection relationships. To this end, analyzing species interactions is crucial for a mechanistic understanding of the factors governing emerging infectious diseases (*Johnson et al., 2015a*).

Here, we provide a comprehensive analysis combining fields of community ecology and virology to understand how viral richness depends on host richness, and which role host–habitat associations play for viral abundance patterns using viruses that naturally infect mosquitoes in a natural system. Habitat alterations such as agricultural development and urbanization promote thriving of disease-transmitting mosquito species (*Burkett-Cadena and Vittor, 2018*; *Lee et al., 2020*; *Perrin et al., 2022*; *Perrin et al., 2023*; *Schrama et al., 2020*), but it is less clear how such changes in mosquito species disassembly affect the abundance and prevalence of viruses. From the limited amount of available studies, these mostly focused on the effects of land use change on one specific virus (*Ferraguti et al., 2021*; *Gardner et al., 2014*; *Martínez-de la Puente et al., 2018*). In addition to the vertebrate-pathogenic arthropod-borne viruses (arboviruses), for example, dengue virus (DENV), yellow fever virus (YFV), and Zika virus (ZIKV) (*Gould et al., 2017*; *Weaver et al., 2018a*), mosquitoes can also be infected with so-called insect-specific viruses (ISVs), which cannot infect vertebrates due to several host restriction barriers, for example, sensitivity to higher (body) temperatures and inability

to replicate in vertebrate cells (*Blitvich and Firth, 2015*; *Marklewitz et al., 2015*). All arbovirus taxa are embedded in a much greater phylogenetic diversity of ISVs, which suggests that arboviruses evolved from ancestral arthropod-restricted viruses (*Blitvich and Firth, 2015*; *Marklewitz et al., 2015*; *Halbach et al., 2017*; *Junglen, 2016*). The mode of transmission of most ISVs in nature remains largely unknown but is suggested to rely on direct transmission. For some insect-specific flaviviruses, vertical transmission to the progeny was described and is considered to play an important role for virus maintenance in nature (*Blitvich and Firth, 2015*; *Bolling et al., 2012*; *Cook et al., 2006*; *Luto-miah et al., 2007*). Additional transmission mechanisms of ISVs may include shared food sources, ectoparasites, or venereal transmission (reviewed in *Halbach et al., 2017*; *Agboli et al., 2019*; *Meki et al., 2021*). It has been shown that at least some flavi-, mesoni, and bunyaviruses are expectorated during feeding, which may provide a route for horizontal virus transmission (*Birnberg et al., 2020*; *Gaye et al., 2020*; *Newton et al., 2020*; *Ramírez et al., 2018*). However, transmission dynamics may differ among virus species and viral families and data are lacking describing the maintenance and transmission of ISVs related to arboviruses (*Meki et al., 2021*). Knowledge of the natural transmission of ISVs has been mainly studied for entomopoxviruses and baculoviruses, which infect other insects than mosquitoes (*Ma et al., 2021*). Due to their high abundance in natural mosquito populations, ISVs can serve as model systems to study how changes in mosquito species assembly affect associated virus abundance.

In a preliminary analysis, we had demonstrated that the prevalence of three ISVs, isolated from mosquitoes sampled along an anthropogenic disturbance gradient in Côte d'Ivoire, West Africa, increased from pristine to disturbed habitat types (*Junglen et al., 2009b*; *Zirkel et al., 2011*). Subsequently, a plenitude of previously unknown RNA viruses was identified in these samples establishing novel species, genera, and even families (*Marklewitz et al., 2015*; *Zirkel et al., 2011*; *Hermanns et al., 2017*; *Hermanns et al., 2014*; *Junglen et al., 2009a*; *Junglen et al., 2017*; *Kallies et al., 2014*; *Marklewitz et al., 2011*; *Marklewitz et al., 2013*; *Quan et al., 2010*; *Zirkel et al., 2013*; *Schuster et al., 2014*). To this end, we established broad-range generic PCR assays for all viruses identified previously in these samples, as well as for all major virus taxa containing arthropod-associated viruses. With this, we aimed to assess the viral genetic diversity along with the prevalence patterns of each detected virus in an entire family of hosts (*Culicidae*) sampled across habitat types. We expect a higher species diversity in mosquitoes of the family *Culicidae* in habitats of low or no disturbance and predict a turnover in mosquito species composition along the disturbance gradient concomitant with an increase in disturbance-resilient mosquito species and their viruses. We thus hypothesize that habitat perturbation and subsequent changes in community composition influence virus abundance patterns and most likely transmission dynamics. This multi-host and multi-taxa study provides insight into common and distinct micro-evolutionary patterns of virus emergence and geographic spread at the interface of pristine and modified landscapes.

## Results

### Biodiversity analyses

The data set included 42 unique mosquito species units. The abundance of different mosquito genera varied considerably across the different habitats, as described previously by *Junglen et al., 2009b*. The dominant genus across all habitats except in the primary forest was *Culex* (50.5%), whereas in the primary forest mosquitoes of the genus *Uranotaenia* were most frequently sampled. *Culex decens* was most abundant in the secondary forest and agricultural areas, whereas *Culex nebulosus* mosquitoes were the most abundant species in villages and at camp sites located in the primary forest (*Figure 1*). A large fraction of 40, 40, and 26% of the sampled mosquitoes in the primary and secondary forest as well as at the camp sites, respectively, could not be identified to species level either due to morphological damage or to limitations of taxonomic keys. The actual number of different species in these habitats is therefore likely to be higher and the relative abundance of some mosquito species may have been underestimated (*Figure 1—figure supplement 1*). We counted and rarefied ($r$ given as asymptotic diversity estimate) the highest number of species ($n$) in the primary forest (PF, $n = 20$ | $r = 38$ | with $i$ number of individuals = 462), followed by camp (C, $n = 18$ | $r = 24$ | $i = 418$) and secondary forest (SF, $n = 14$ | $r = 24$ | $i = 651$), agriculture (A, $n = 17$ | $r = 18$ | $i = 882$), and village (V, $n = 13$ | $r = 15$ | $i = 857$) (*Figure 1—figure supplement 1*).

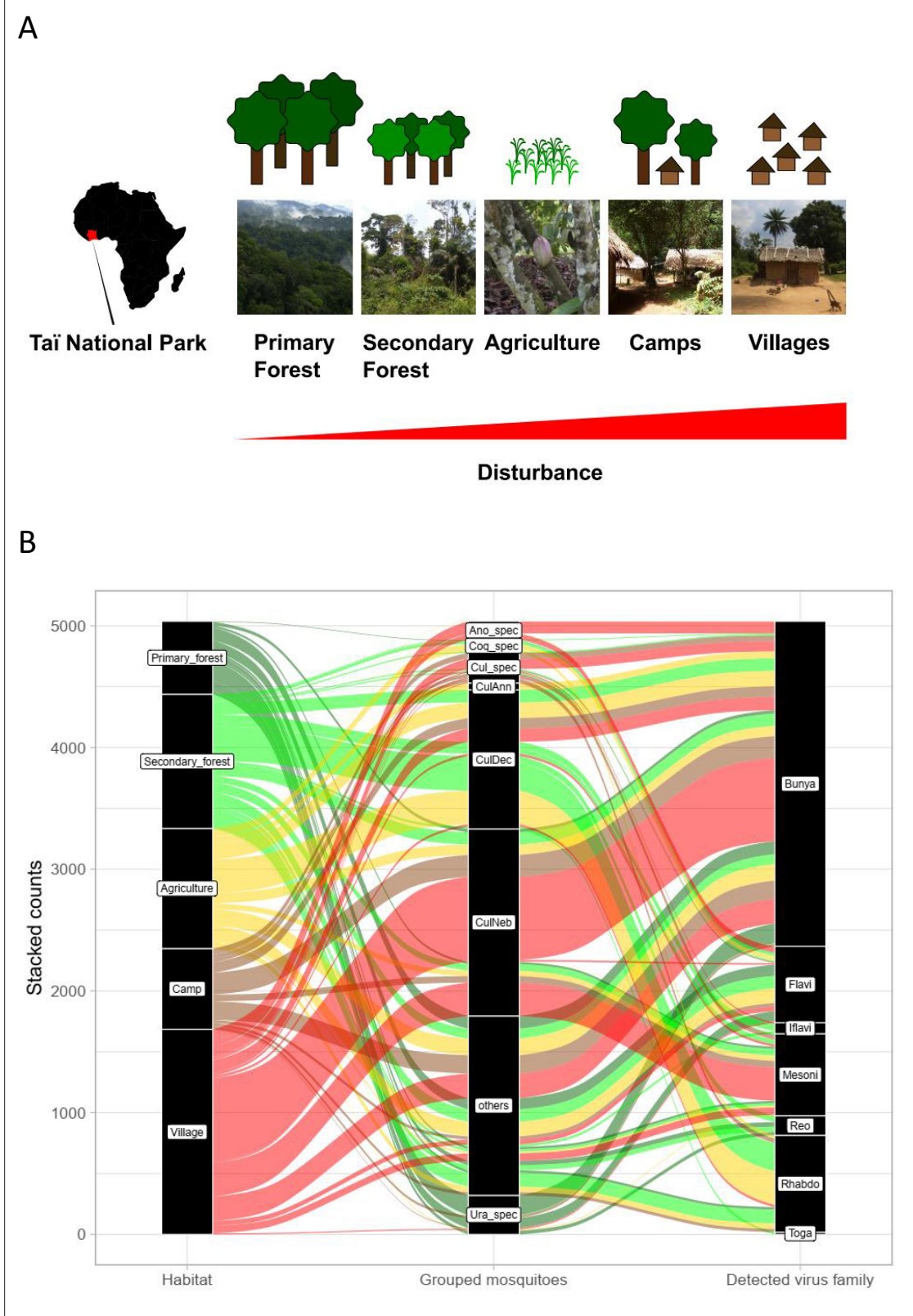

**Figure 1.** Study sites and mosquito distribution. (**A**) Study site location and overview over the five habitat types. The different habitat types along the anthropogenic disturbance gradient are depicted by photos and drawings. (**B**) Alluvial plot showing the distribution of the main mosquito species groups and main virus families to the five respective habitats. CulAnn: *Culex annulioris*; CulDec: *Culex decens;* CulNeb: *Culex nebulosus;* Cul_spec: other *Culex* species; Ano_spec: *Anopheles* species; Ura_spec: *Uranotaenia* species; Coq_spec: *Coquillettidia* species; others: all other grouped species

*Figure 1 continued on next page*

*Figure 1 continued*

(see main text). Bunya: order *Bunyavirales* containing *Phenuiviridae*, *Peribunyaviridae*, and *Phasmaviridae*; Flavi: *Flaviviridae*; Toga: *Togaviridae*; Rhabdo: *Rhabdoviridae*; Reo: *Reoviridae*; Iflavi: *Iflaviridae*; Meso: *Mesoniviridae*.

The online version of this article includes the following figure supplement(s) for figure 1:

**Figure supplement 1.** Biodiversity analyses.

**Figure supplement 2.** Hierarchical cluster analysis based on associations between mosquito species. Main mosquito groups were identified as CulAnn, CulDec, CulNeb, Cul_spec, Ura_spec, Ano_spec, and Coq_spec. CulAnn: *Culex annulioris*; CulDec: *Culex decens*; CulNeb: *Culex nebulosus*; Cul_spec: other *Culex species*; Ano_spec: *Anopheles species*; Ura_spec: *Uranotaenia species*; Coq_spec: *Coquillettidia species*, others: all other grouped species. PF: primary forest; SF: secondary forest; A: agriculture; C: camp; V: village.

Rarefaction curves were still increasing for camp, primary and secondary forest, indicating that a higher trapping effort would have led to a higher number of species. However, confidence intervals were strongly overlapping, especially for camp and primary forest and for agriculture, secondary forest, and village. Bray–Curtis dissimilarity (BCd) yielded the highest difference in community assembly between primary forest and village (BCd = 0.90), and highest similarity between agriculture and secondary forest (BCd = 0.69). The hierarchical cluster analysis confirmed the finding that agriculture and secondary forest clustered together via *C. decens* and *C. nebulosus*, while villages and camp mainly clustered via *C. nebulosus* (*Figure 1—figure supplement 2*). Agriculture had high amounts of *Culex annulioris* and *Coquillettidia metallica*, while primary forest mosquito communities were characterized by *Uranotaenia* species. Villages were mainly characterized by *Anopheles* species (*Figure 1—figure supplement 2*). We therefore grouped our mosquitoes to the eight major groups *C. decens* (i = 1101), *C. nebulosus* (i = 719), *C. annulioris* (i = 263), *Culex* sp. (i = 222), *Coquillettidia* sp. (i = 95), *Uranotaenia* sp. (i = 502), and *Anopheles* sp. (i = 294), while all others, mainly the non-defined ones, were termed 'others' (i = 1366) (*Figure 1B*). Based on these findings, our anthropogenic disturbance gradient was as follows (from low to high disturbance): PF-SF-A-C-V. Significant associations were found between main mosquito groups and habitat (LRT p<0.001, df = 28, deviance = 3782; *Supplementary file 1* and *Figure 1B*).

## Assessment of the genetic virus diversity

In total, we found 331 viral RdRp sequences pertaining to 34 putative novel viruses and 15 previously identified viruses of the families *Phenuiviridae*, *Peribunyaviridae*, and *Phasmaviridae* of the order *Bunyavirales*, as well as of the families *Flaviviridae*, *Togaviridae*, *Rhabdoviridae*, *Reoviridae*, *Iflaviridae*, and *Mesoniviridae* (*Table 1*). Sequences with at least 5% pairwise amino acid distance to known viral RdRp sequences were suggested to pertain to distinct viral species.

The family *Phenuiviridae* (order *Bunyavirales*) includes important arboviruses within the genus *Phlebovirus* but also numerous ISVs that for example belong to the genera *Goukovirus* and *Phasivirus* (*Abudurexiti et al., 2019*). Nine distinct phenuiviruses were detected, which included seven novel viruses, named Sefomo virus (acronym for *se*condary *fo*rest *mo*squito virus) and Cimo phenuivirus I–VI, as well as Gouléako virus (GOLV) and Phasi Charoen-like virus (PCLV) (*Marklewitz et al., 2011*; *Chandler et al., 2014*). For all viruses, the number of positive samples per habitat is summarized in *Table 1*. Phylogenetic analyses showed that the viruses grouped with ISVs of the genera *Goukovirus*, *Phasivirus*, *Hudivirus*, and *Beidivirus*, as well as with the unclassified insect viruses related to the uncultured virus isolate acc 9.4 (*Figure 2A*). Interestingly, Cimo phenuivirus V branched basal to tenuiviruses that are transmitted between plants by planthoppers (*Nault and Ammar, 1989*). Cimo phenuivirus V was found in *Anopheles* spp. mosquitoes collected at a rice plantation. At this point we cannot differentiate whether the mosquito ingested infected plant material or whether this novel tenui-like virus can infect mosquitoes.

The family *Peribunyaviridae* (order *Bunyavirales*) consists of two arbovirus genera (*Orthobunyavirus* and *Pacuvirus*) and two genera that contain ISVs (*Herbevirus* and *Shangavirus*) (*Hughes et al., 2020*). We identified two novel peribunyaviruses, named Cimo peribunyavirus I and II. These viruses formed a monophyletic clade that shared a most recent common ancestor with arboviruses of the genera *Orthobunyavirus* and *Pacuvirus* (*Figure 2B*). In addition, the insect-specific herbeviruses Taï virus and Herbert virus (HEBV) (*Marklewitz et al., 2013*), as well as a previously undescribed herbevirus, named Cimo peribunyavirus III, were detected that fell into the clade of herbeviruses (*Figure 2B*).

**Table 1.** Distribution and host association of detected viruses.

Number of positive pools per habitat and mosquito host species of all detected viruses and virus-like sequences. The main mosquito host species are indicated in bold letters. PF: primary forest; SF: secondary forest; A: agriculture; C: camp; V: village; nd: not determined.

| Virus family | Virus | No. of positive pools | | | | | | Live virus isolate | Mosquito species | Representative pool (GenBank accession number)* | Sequence length (nt)† | References |
|---|---|---|---|---|---|---|---|---|---|---|---|---|
| | | Σ | PF | SF | A | C | V | | | | | |
| *Phenuiviridae* | Gouléako virus | 33 | 2 | 4 | 5 | 6 | 16 | × | *Culex nebulosus, Culex decens, Culex* spp., nd | A05 (NC_043051) | Full genome | *Marklewitz et al., 2011* |
| | Cimo phenuivirus I | 1 | 0 | 0 | 0 | 1 | 0 | | nd | A27 (MZ202291) | 892 | This study |
| | Cimo phenuivirus II | 12 | 8 | 1 | 0 | 3 | 0 | | *Uranotaenia mashonaensis, Uranotaenia ornata, Uranotaenia* spp., nd | B02 (MZ202292) | 1096 | This study |
| | Cimo phenuivirus III | 2 | 2 | 0 | 0 | 0 | 0 | | nd | B14 (MZ202293) | 1099 | This study |
| | Cimo phenuivirus IV | 1 | 1 | 0 | 0 | 0 | 0 | | nd | B98 (MZ202294) | 1096 | This study |
| | Cimo phenuivirus V | 1 | 0 | 0 | 1 | 0 | 0 | | *Anopheles* spp. | E02 (MZ202298) | 1091 | This study |
| | Cimo phenuivirus VI | 6 | 0 | 0 | 0 | 0 | 6 | | *Anopheles gambiae, Anopheles nili, Anopheles* spp., nd | F09 (MZ202300) | 1210 | This study |
| | Sefomo virus | 1 | 0 | 1 | 0 | 0 | 0 | × | *Culex decens* | C43 (MZ202295) | Full genome | This study |
| | Phasi Charoen-like virus | 2 | 0 | 0 | 1 | 0 | 1 | | *Aedes aegypti*, nd | F04 (MZ202299) | 1092 | This study |

*Table 1 continued on next page*

Table 1 continued

| Virus family | Virus | No. of positive pools | | | | | | Mosquito species | Live virus isolate | Representative pool (GenBank accession number)* | Sequence length (nt)† | References |
|---|---|---|---|---|---|---|---|---|---|---|---|---|
| | | Σ | PF | SF | A | C | V | | | | | |
| Peribunyaviridae | Herbert virus | 42 | 3 | 8 | 8 | 6 | 17 | Culex nebulosus, Culex decens, Culex spp., Mimomyia mimomyiaformis, Coquillettidia spp., nd | × | F23 (NC_038714) | Full genome | Marklewitz et al., 2013 |
| | Taï virus | 3 | 0 | 2 | 1 | 0 | 0 | Culex (Culex) decens, nd | × | F47 (NC_034459) | Full genome | Marklewitz et al., 2013 |
| | Cimo peribunyavirus I | 4 | 3 | 0 | 1 | 0 | 0 | Uranotaenia mashonaensis, nd | | B04 (MZ202287) | 2278 | This study |
| | Cimo peribunyavirus II | 1 | 0 | 0 | 1 | 0 | 0 | nd | | D55 (MZ202289) | 3596; 2228 (M segment) | This study |
| | Cimo peribunyavirus III | 2 | 0 | 1 | 0 | 1 | 0 | Culex nebulosus, Culex decens | | A07 (MZ202286) | 1488 | This study |
| Phasmaviridae | Ferak virus | 20 | 1 | 3 | 4 | 6 | 6 | Culex nebulosus, Culex decens, nd | × | C51 (NC_043031) | Full genome | Marklewitz et al., 2015 |
| | Jonchet virus | 17 | 2 | 0 | 0 | 9 | 6 | Culex decens, Culex nebulosus, Culex spp., nd | × | B81 (NC_038706) | Full genome | Marklewitz et al., 2015 |
| | Spilikins virus | 12 | 1 | 2 | 2 | 2 | 5 | Culex nebulosus, Culex spp., nd | | A28 (MZ202269) | Complete CDS | This study |
| | Mikado virus | 3 | 0 | 0 | 3 | 0 | 0 | Culex annulioris | × | D35 (MZ202272) | Complete CDS | This study |

Table 1 continued on next page

*Table 1 continued*

| Virus family | Virus | No. of positive pools | | | | | | Mosquito species | Live virus isolate | Representative pool (GenBank accession number)* | Sequence length (nt)† | References |
|---|---|---|---|---|---|---|---|---|---|---|---|---|
| | | Σ | PF | SF | A | C | V | | | | | |
| *Rhabdoviridae* | Cimo rhabdovirus I | 40 | 0 | 20 | 15 | 3 | 2 | *Culex decens, Culex* spp., nd | | A02 (MZ202301) | 1146 | This study |
| | Cimo rhabdovirus II | 1 | 0 | 0 | 0 | 1 | 0 | nd | | A30 (MZ202302) | 988 | This study |
| | Cimo rhabdovirus III | 2 | 2 | 0 | 0 | 0 | 0 | nd | | B58 (MZ202303) | 769 | This study |
| | Cimo rhabdovirus IV | 2 | 0 | 1 | 1 | 0 | 0 | nd | | C68 (MZ202304) | 980 | This study |
| | Cimo rhabdovirus V | 3 | 0 | 0 | 3 | 0 | 0 | *Coquillettidia metallica*, nd | | D24 (MZ202305) | 898 | This study |
| *Potential Rhabdovirus-like NIRVS* | *Rhabdovirus-like NIRVS I* | 1 | 0 | 0 | 0 | 1 | 0 | nd | | A19 (MZ399708) | 638 | This study |
| | *Rhabdovirus-like NIRVS II* | 1 | 1 | 0 | 0 | 0 | 0 | nd | | B82 (MZ399709) | 566 | This study |
| *Reoviridae* | Cimodo virus | 5 | 0 | 5 | 0 | 0 | 0 | *Culex decens*, nd | × | C74 (NC_023420) | Full genome | *Hermanns et al., 2014* |
| | Wanken orbivirus | 5 | 5 | 0 | 0 | 0 | 0 | *Uranotaenia mashonaensis*, nd | | B14 (MZ202276) | Full genome | This study |
| *Mesoniviridae* | Cavally virus | 30 | 3 | 5 | 4 | 3 | 15 | *Culex nebulosus, Culex decens, Culex* spp., nd | × | C79 (NC_015668) | Full genome | *Zirkel et al., 2011* |
| | Hana virus | 1 | 0 | 0 | 0 | 1 | 0 | *Culex* spp. | × | A04 (NC_020899) | Full genome | *Zirkel et al., 2013* |
| | Méno virus | 1 | 0 | 0 | 1 | 0 | 0 | *Uranotaenia* spp. | × | E09 (NC_020900) | Full genome | *Zirkel et al., 2013* |
| | Nsé virus | 7 | 1 | 3 | 1 | 1 | 1 | *Culex nebulosus, Culex decens* | × | F24 (NC_020901) | Full genome | *Zirkel et al., 2013* |
| *Togaviridae* | Taï Forest alphavirus | 1 | 0 | 1 | 0 | 0 | 0 | *Culex decens* | | C21 (NC_032681) | Full genome | *Hermanns et al., 2017* |

*Table 1 continued on next page*

*Table 1 continued*

| Virus family | Virus | No. of positive pools | | | | | | | Mosquito species | Live virus isolate | Representative pool (GenBank accession number)* | Sequence length (nt)† | References |
|---|---|---|---|---|---|---|---|---|---|---|---|---|---|
| | | Σ | PF | SF | A | C | V | | | | | |
| *Iflaviridae* | Cimo iflavirus I | 2 | 0 | 1 | 1 | 0 | 0 | *Culex decens, Culex nebulosus* | | D52 (MZ202268) | 1105 | This study |
| | Cimo iflavirus II | 1 | 0 | 1 | 0 | 0 | 0 | *Culex decens* | | C61 (MZ202265) | 186 | This study |
| | Cimo iflavirus III | 2 | 0 | 2 | 0 | 0 | 0 | *Culex decens* | | C95 (MZ202267) | 730 | This study |
| | Sassandra virus | 2 | 0 | 1 | 0 | 0 | 1 | *Culex* spp., nd | × | C93 (MZ202266) | Full genome | This study |

*Table 1 continued*

| Virus family | Virus | No. of positive pools | | | | | | Mosquito species | Live virus isolate | Representative pool (GenBank accession number)* | Sequence length (nt)† | References |
|---|---|---|---|---|---|---|---|---|---|---|---|---|
| | | Σ | PF | SF | A | C | V | | | | | |
| *Flaviviridae* | Niénokoué virus | 9 | 2 | 3 | 3 | 0 | 1 | *Coquillettidia metallica*, *Culex* spp., nd | × | B51 (NC_024299) | Full genome | *Junglen et al., 2017* |
| | Nounané virus | 3 | 3 | 0 | 0 | 0 | 0 | *Uranotaenia mashonaensis* | × | B31 (NC_033715) | Complete CDS | *Junglen et al., 2009a* |
| | Anopheles flavivirus | 3 | 0 | 0 | 0 | 1 | 2 | *Anopheles gambiae*, *Anopheles* spp. | | F10 (MZ202263) | 1172 | This study |
| | Tafomo virus | 7 | 2 | 1 | 2 | 2 | 0 | *Culex* spp., nd | × | B22 (MZ202252) | Full genome | This study |
| | Cimo flavivirus I | 13 | 4 | 6 | 3 | 0 | 0 | *Coquillettidia* spp. (unknown COI-type C69), nd | | B36 (MZ202253) | 702 | This study |
| | Cimo flavivirus II | 4 | 1 | 0 | 3 | 0 | 0 | *Uranotaenia* spp., nd | | D01 (MZ202258) | 790 | This study |
| | Cimo flavivirus III | 5 | 2 | 0 | 0 | 2 | 1 | *Uranotaenia mashonaensis*, *Uranotaenia* spp., nd | | B01 (MZ202251) | 790 | This study |
| | Cimo flavivirus IV | 5 | 0 | 0 | 4 | 0 | 1 | *Mimomyia* spp., nd | | E08 (MZ202261) | 514 | This study |
| | Cimo flavivirus V | 3 | 0 | 0 | 2 | 0 | 1 | *Mimomyia hispida*, nd | | D20 (MZ202259) | 1192 | This study |
| | Cimo flavivirus VI | 3 | 0 | 0 | 2 | 0 | 1 | *Anopheles rhodesiensis rupicolus*, *Anopheles* spp. | | E02 (MZ202260) | 516 | This study |
| | Cimo flavivirus VII | 3 | 1 | 2 | 0 | 0 | 0 | nd | | B36 (MZ202254) | 1188 | This study |
| | Cimo flavivirus VIII | 2 | 1 | 1 | 0 | 0 | 0 | *Eretmapodites intermedius*, nd | | B85 (MZ202255) | 1188 | This study |
| | Cimo flavivirus IX | 1 | 0 | 1 | 0 | 0 | 0 | nd | | C51 (MZ202257) | 1188 | This study |
| | Cimo flavivirus X | 1 | 0 | 0 | 1 | 0 | 0 | nd | | E08 (MZ202262) | 736 | This study |
| | Cimo flavivirus XI | 1 | 0 | 0 | 0 | 0 | 1 | *Culex nebulosus* | | F41 (MZ202264) | 1193 | This study |

*Table 1 continued*

| Virus family | Virus | No. of positive pools | | | | | | Mosquito species | Live virus isolate | Representative pool (GenBank accession number)* | Sequence length (nt)† | References |
|---|---|---|---|---|---|---|---|---|---|---|---|---|
| | | Σ | PF | SF | A | C | V | | | | | |
| Potential Flavivirus-like NIRVS | Flavivirus-like NIRVS I | 10 | 2 | 3 | 4 | 0 | 1 | *Coquillettidia metallica*, nd | | B60 (MZ399699) | 943 | This study |
| | Flavivirus-like NIRVS II | 2 | 0 | 0 | 1 | 0 | 1 | *Aedes aegypti*, *Aedes* spp. | | E01 (MZ399703) | 846 | This study |
| | Flavivirus-like NIRVS III | 3 | 0 | 2 | 0 | 1 | 0 | *Eretmapodites* spp., nd | | C78 (MZ399701) | 668 | This study |
| | Flavivirus-like NIRVS IV | 2 | 2 | 0 | 0 | 0 | 0 | nd | | B18 (MZ399698) | 515 | This study |
| | Flavivirus-like NIRVS V | 2 | 1 | 0 | 0 | 1 | 0 | nd | | B68 (MZ399700) | 512 | This study |
| | Flavivirus-like NIRVS VI | 2 | 0 | 0 | 0 | 0 | 2 | *Mimomyia* spp., nd | | F16 (MZ399707) | 383 | This study |
| | Flavivirus-like NIRVS VII | 3 | 0 | 2 | 1 | 0 | 0 | nd | | D57 (MZ399702) | 332 | This study |
| | Flavivirus-like NIRVS VIII | 13 | 8 | 1 | 2 | 2 | 0 | *Uranotaenia* spp., nd | | A25 (MZ399697) | 835 | This study |
| | Flavivirus-like NIRVS IX | 8 | 2 | 1 | 2 | 2 | 1 | *Uranotaenia* spp., *Uranotaenia mashonaensis*, nd | | E26 (MZ399706) | 650 | This study |
| | Flavivirus-like NIRVS X | 15 | 4 | 3 | 6 | 1 | 1 | *Coquillettidia metallica*, nd | | E11 (MZ399705) | 640 | This study |

*RdRp encoding sequence (previously published sequences are indicated in italic).

†RdRp encoding segment or sequence, unless otherwise stated.

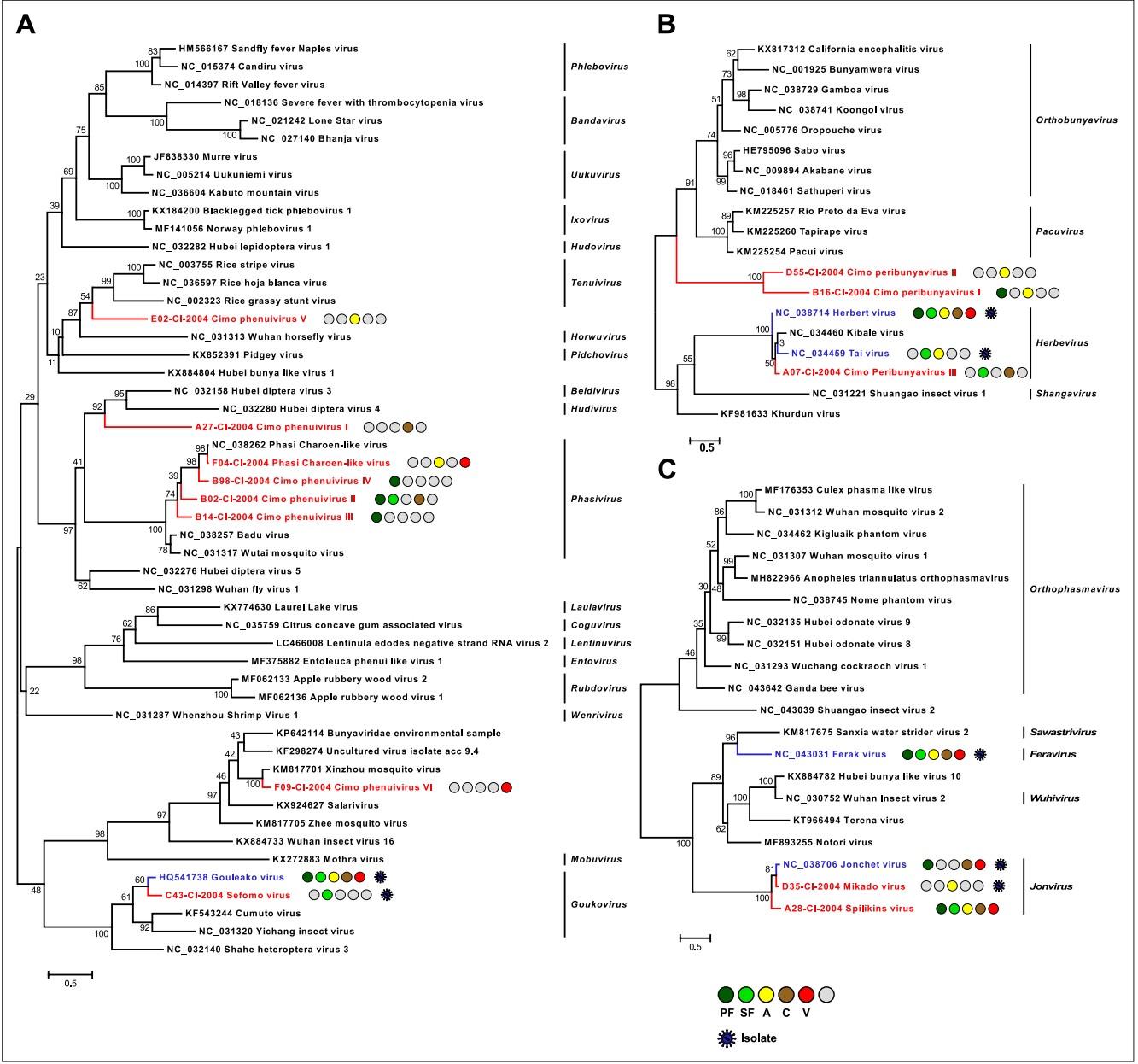

**Figure 2.** Phylogenetic analyses of detected bunyaviruses. Phylogenetic trees were inferred with PhyML (LG substitution model) based on MAFFT-E protein alignments covering the conserved RdRp motifs of the families *Phenuiviridae* (**A**), *Peribunyaviridae* (**B**), and *Phasmaviridae* (**C**). The alignment length was approximately 290, 300, and 690 amino acids, respectively. Novel viruses from this study are indicated in red, and previously published viruses detected in our data set are indicated in blue. Sample origin from the different habitat types is indicated by colored circles while no detection is indicated by gray circles. Live virus isolates are marked with a blue virion. PF: primary forest; SF: secondary forest; A: agriculture; C: camp; V: village.

The online version of this article includes the following figure supplement(s) for figure 2:

**Figure supplement 1.** Phylogenetic analyses of detected phasmaviruses.

The family *Phasmaviridae* (order *Bunyavirales*) comprises only ISVs (*Abudurexiti et al., 2019*). We found two prototype species of the genera *Feravirus* and *Jonvirus*, Ferak virus (FERV) and Jonchet virus (JONV), which were previously isolated in cell culture from these mosquitoes (*Marklewitz et al., 2015*). While a great diversity of novel phasmaviruses has been found since the first discovery of this family, no additional members of the genus *Jonvirus* have been identified (*Abudurexiti et al., 2019*). Here, we further detected two previously unknown jonviruses, named Mikado virus and Spilikins virus, which are closely related to the prototype virus JONV in phylogenetic analyses (*Figure 2C*). For both

novel jonviruses, the complete coding sequence (CDS) was sequenced and phylogenies of the M and S segments are shown in *Figure 2—figure supplement 1*. In all phylogenies, the two novel viruses cluster are well supported with JONV.

The family *Rhabdoviridae* (order *Mononegavirales*) is highly diversified and currently contains 20 genera. Rhabdoviruses infect vertebrates, arthropods, and plants (*Walker et al., 2018*). We

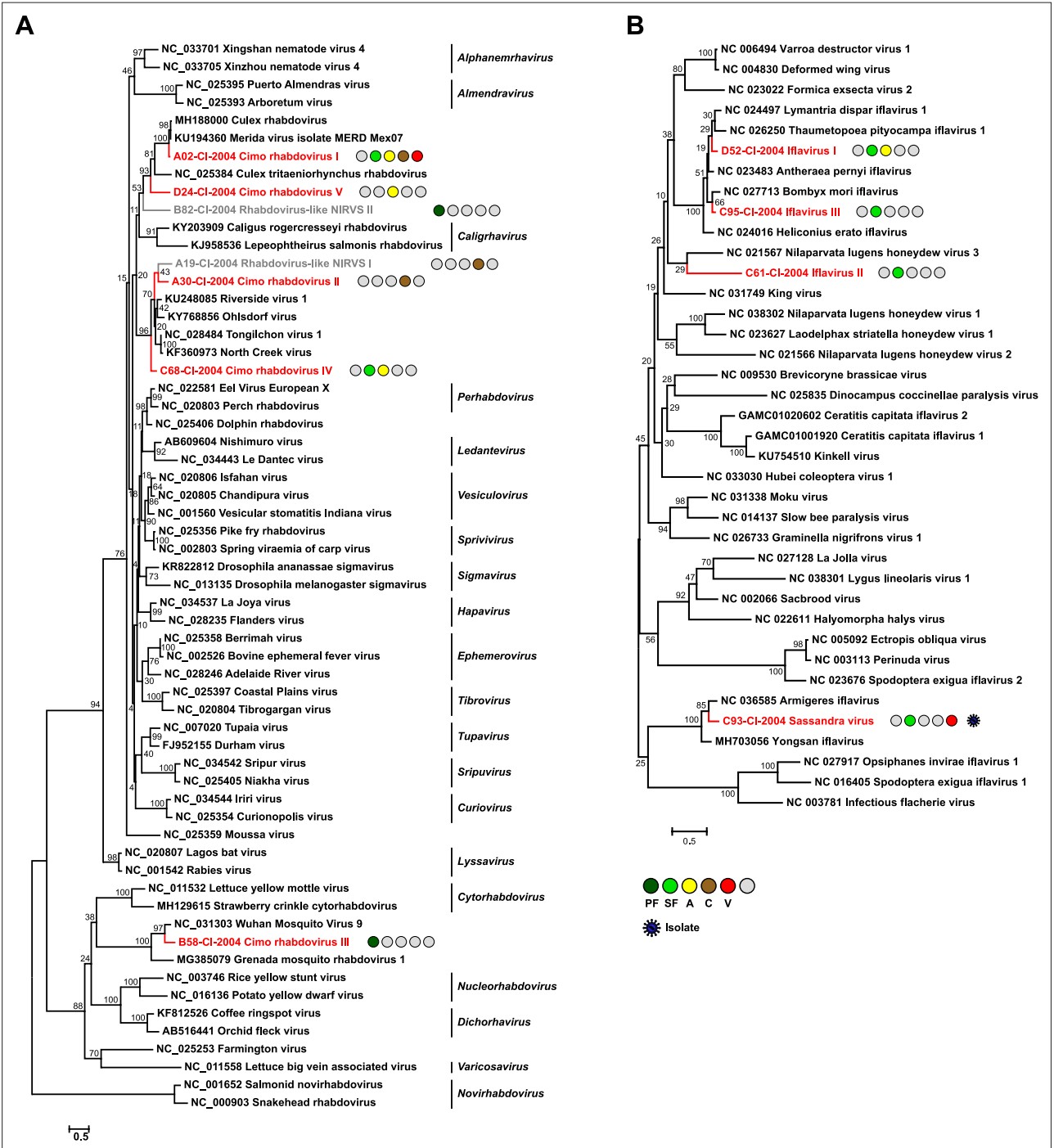

**Figure 3.** Phylogenetic analyses of detected rhabdoviruses and iflaviruses. Phylogenetic trees were inferred with PhyML (LG substitution model) based on MAFFT-E protein alignments covering the conserved RdRp motifs of the families *Rhabdoviridae* (**A**) and *Iflaviridae* (**B**). The alignment length was approximately 270 and 250 amino acids, respectively. Novel viruses from this study are indicated in red, and detected virus-like sequences are indicated in gray. Sample origin from the different habitat types is indicated by colored circles while no detection is indicated by gray circles. Live virus isolates are marked with a blue virion. PF: primary forest; SF: secondary forest; A: agriculture; C: camp: V: village.

detected five previously unknown rhabdoviruses, named Cimo rhabdovirus I–V, that clustered with different unclassified clades of mosquito-associated rhabdoviruses across the rhabdovirus phylogeny (*Figure 3A*).

The family *Iflaviridae* (order *Picornavirales*) consists of a single genus whose members are restricted to arthropod hosts (*Valles et al., 2017*). We detected four novel iflaviruses, named Sassandra virus and Cimo iflavirus I–III, which were placed in three different clades in phylogenetic analyses (*Figure 3B*). Cimo iflavirus I and III grouped with Bombyx mori iflavirus and other lepidopteran iflaviruses, while the short sequence fragment of Cimo iflavirus II did not form a well-supported clade with known iflaviruses. Sassandra virus clustered with two previously described iflaviruses from mosquitoes (*Figure 3B*).

The genus *Flavivirus* (family *Flaviviridae*) includes important arboviruses as well as viruses with a single host tropism for arthropods or vertebrates (*Zell et al., 2017*). Insect-specific flaviviruses can be divided into two groups. Classical insect-specific flaviviruses form a monophyletic clade in basal phylogenetic relationship to all other flaviviruses while dual-host-affiliated insect-specific flaviviruses are phylogenetically affiliated with the arboviruses of this genus (*Blitvich and Firth, 2015*). We found 12 undescribed flaviviruses, named Tafomo virus (acronym for *Taï forest mosquito virus*) and Cimo flavivirus I–XI, as well as a strain of Anopheles flavivirus (*Fauver et al., 2016*). All sequences clustered within the clade comprising the classical insect-specific flaviviruses in phylogenetic analysis (*Figure 4A*). In addition, two previously characterized flaviviruses, the dual-host-affiliated insect-specific flavivirus Nounané virus and the classical insect-specific flavivirus Niénokoué virus (NIEV), were detected in the mosquitoes (*Junglen et al., 2009a*; *Junglen et al., 2017*).

The family *Reoviridae* comprises 15 genera with highly variable biological properties. Three genera contain arboviruses (*Orbivirus*, *Coltivirus,* and *Seadornavirus*) while viruses belonging to other genera infect vertebrates, plants, fungi, or insects (*Becnel, 2012*). One novel orbivirus, named Wanken orbivirus (WKOV), was detected in the mosquitoes. The complete genome of WKOV was sequenced and the polymerase encoded on the first segment showed 56% pairwise amino acid identity to Ninarumi virus. Phylogenetic analysis of the polymerase (VP1) placed WKOV on a branch with Ninarumi virus, a virus detected in *Aedes fulvus* mosquitoes collected in Peru (*Sadeghi et al., 2017*). This clade was placed basal to the clade of *Culicoides*- and *Phlebotominae*-transmitted orbiviruses (e.g. Bluetongue virus) and Letea virus, a virus detected in snakes (*Natrix natrix*) captured in Romania (*Tomazatos et al., 2020*; *Figure 4B*). Additional phylogenetic analyses based on VP3, VP4, VP5, und VP7 also placed WKOV together with Ninarumi virus and Letea virus basal to the clade of *Culicoides*- and *Phlebotominae*-transmitted orbiviruses (*Figure 4—figure supplement 1*). Additionally, Cimodo virus, which most likely defines a novel reovirus genus and which was previously characterized (*Hermanns et al., 2014*), was found (*Figure 4—figure supplement 2A*).

Most viruses of the genus *Alphavirus* (family *Togaviridae*) are arboviruses. Contrary to the genus *Flavivirus*, only few insect-specific alphaviruses have been discovered thus far (*Chen et al., 2018*). We detected the previously characterized insect-specific Taï Forest alphavirus (*Hermanns et al., 2017*) but no further alphaviruses were found (*Figure 4—figure supplement 2B*).

The family *Mesoniviridae* (order *Nidovirales*) consists of ISVs (*Zirkel et al., 2013*; *Diagne et al., 2020*). The four previously described mesoniviruses Cavally virus (CAVV), Nsé virus, Hana virus, and Méno virus were detected in the mosquitoes (*Zirkel et al., 2013*; *Figure 4—figure supplement 2C*).

Besides the 12 previously published virus isolates from this sampling (*Marklewitz et al., 2015*; *Zirkel et al., 2011*; *Hermanns et al., 2014*; *Junglen et al., 2009a*; *Junglen et al., 2017*; *Kallies et al., 2014*; *Marklewitz et al., 2011*; *Marklewitz et al., 2013*; *Quan et al., 2010*; *Zirkel et al., 2013*; *Schuster et al., 2014*), Sefomo virus, Mikado virus, Sassandra virus, and Tafomo virus were isolated in C6/36 mosquito cells, indicating the detection of functional viruses. The viruses replicated well in C6/36 cells incubated at 28–30°C but virus growth was impaired at 32°C for all but Mikado virus. No virus could replicate at 34°C, indicating the detection of ISV (*Figure 5*). All other viruses could not be isolated in cell culture.

## Detection of non-retroviral integrated RNA virus sequences

Genome fragments of flavi-, bunya-, and rhabdoviruses were found to have integrated into mosquito genomes and persist as so called non-retroviral integrated RNA virus sequences (NIRVS) (*Palatini et al., 2017*; *Crochu et al., 2004*; *Houé et al., 2019*; *Roiz et al., 2009*). We detected flavi- and rhabdovirus-like sequences with defective ORFs within the conserved region of the RdRp gene,

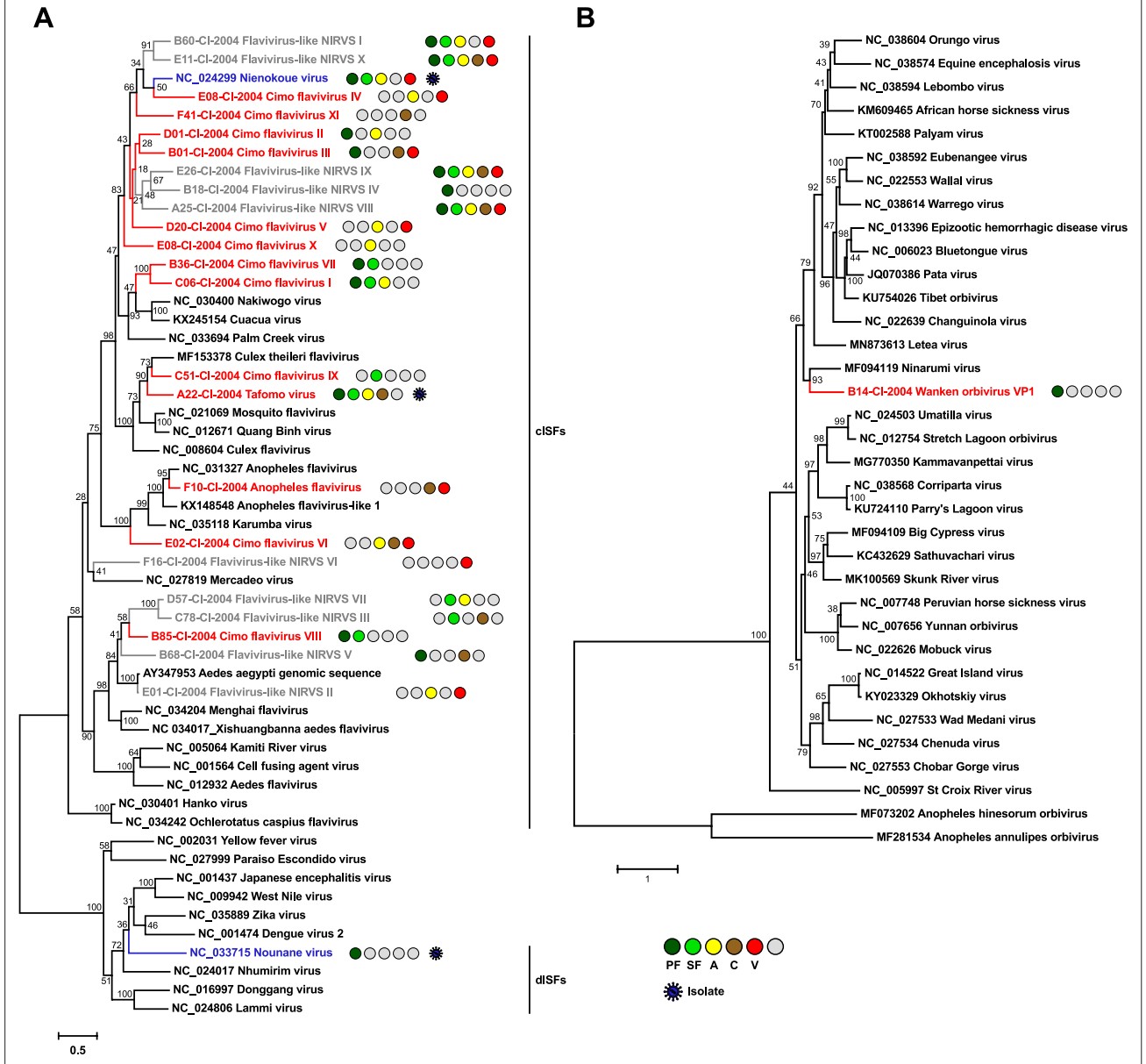

**Figure 4.** Phylogenetic analyses of detected flaviviruses and orbiviruses. Phylogenetic trees were inferred with PhyML (GTR substitution model) based on MAFFT-E nucleotide alignments covering the conserved RdRp motifs of the genus *Flavivirus* (**A**) and with PhyML (LG substitution model) based on a MAFFT-E protein alignment of the polymerase of the genus *Orbivirus* (**B**). The alignment length was approximately 1170 nucleotides and 1190 amino acids, respectively. Novel viruses from this study are indicated in red, previously published viruses detected in our data set are indicated in blue, and detected virus-like sequences are indicated in gray. Sample origin from the different habitat types is indicated by colored circles while no detection is indicated by gray circles. Live virus isolates are marked with a blue virion. PF: primary forest; SF: secondary forest; A: agriculture; C: camp; V: village.

The online version of this article includes the following figure supplement(s) for figure 4:

**Figure supplement 1.** Phylogenetic analyses of the detected Wanken orbivirus (WKOV) and members of the genus *Orbivirus*.

**Figure supplement 2.** Phylogenetic analyses of detected reoviruses, alphaviruses, and mesoniviruses.

**Figure supplement 3.** Potential non-retroviral integrated RNA virus sequences (NIRVS).

suggesting the detection of NIRVS. These findings were not included in the virus diversity and prevalence analyses.

The two potential rhabdovirus-like NIRVS encoded either an internal stop codon (rhabdovirus-like NIRVS II – B82) or sequence elongation attempts resulted in rhabdovirus-like sequences that contained frame shifts (rhabdovirus-like NIRVS I – A19). We were further able to amplify these defective virus-like

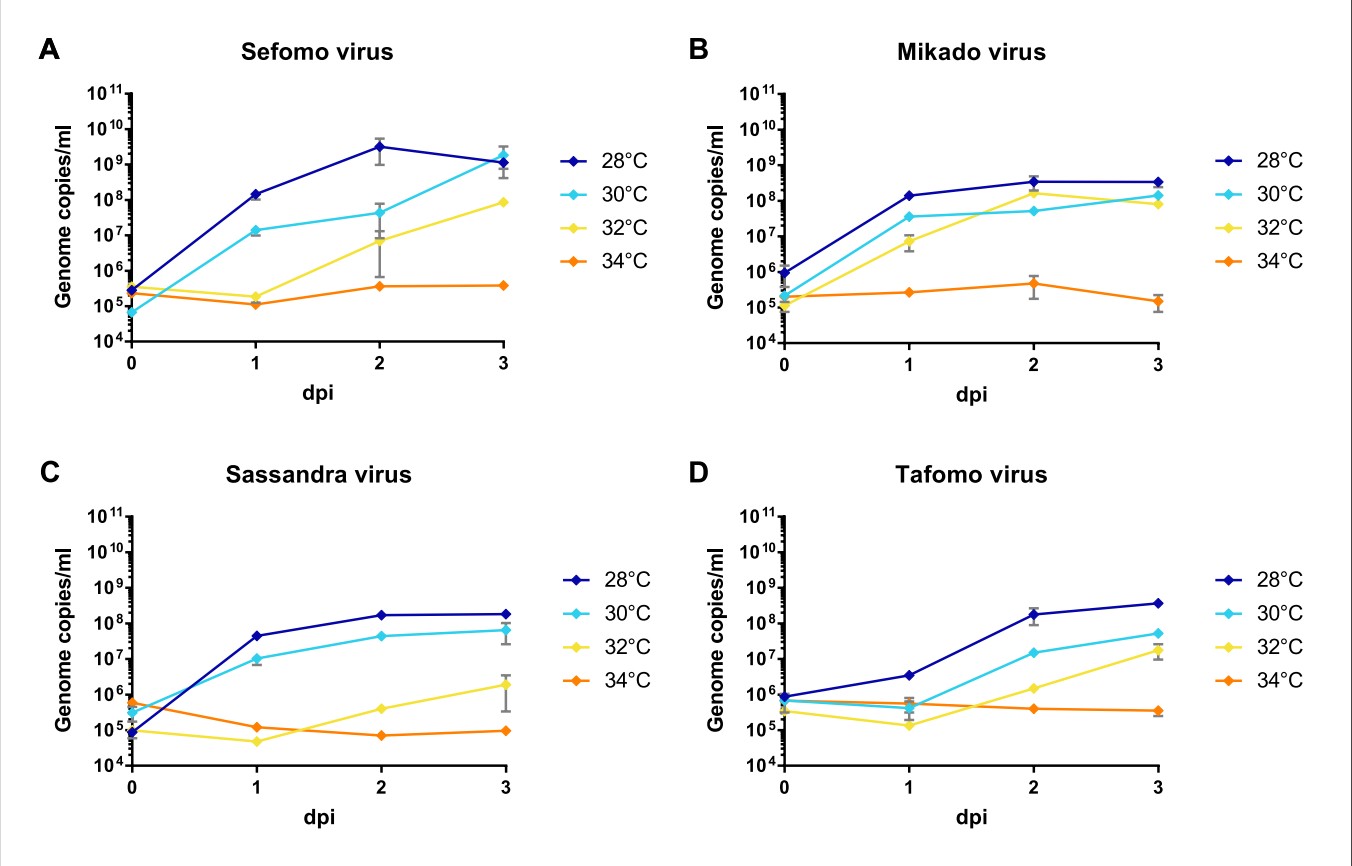

**Figure 5.** Temperature-dependent replication of novel virus isolates. C6/36 cells were infected with an MOI of 0.1 with Sefomo virus (**A**), Mikado virus (**B**), Sassandra virus (**C**), and Tafomo virus (**D**). Replication was measured for 3 dpi at 28, 30, 32, and 34°C.

sequences directly from nucleic acid extracts without prior cDNA synthesis, suggesting that these sequences were present as DNA. No amplificates were obtained for the sequences with contiguous ORFs (*Figure 4—figure supplement 3A*).

Furthermore, we obtained 10 flavivirus-like sequences with frame shifts, internal stop codons, deletions, or integrations (*Figure 4—figure supplement 3B*). In case of flavivirus-like NIRVS II and III, parts of the potentially integrated sequences were related to sequences from insects including genome loci from *Aedes* mosquitoes. Flavivirus-like NIRVS II was amplified from an *Aedes aegypti* pool (E01) and the integrated fragment (68 nt) consisted of 56 nt with 81% identity to *A. aegypti* steroid hormone receptor homolog (AaHR3-2) gene and a 12 nt duplication of the sequence immediately adjacent to the integration site. In addition to the interrupted flavivirus-like sequence in pool E01, an identical continuous flavivirus sequence was obtained from the same pool. A similar observation was made with flavivirus-like NIRVS I (pool B60). The sequence of flavivirus-like NIRVS III (pool C78) was detected in *Eretmapodites intermedius* and undetermined mosquitoes. This sequence was profoundly defective (frameshifts, internal stop codon, and deletions) and continued into a 416-nt-long sequence with low similarity to *Aedes albopictus* and *Apis* spp. genome loci (*Figure 4—figure supplement 3B*). The very few and short sequences of *Eretmapodites* spp. available in GenBank most likely impeded the identification of the host genome sequence of flavivirus-like NIRV III.

## Detected viruses show a high host specificity

We next sought to identify whether the detected viruses were associated with specific mosquito host species. In total, 43% of the pools were found positive for at least one virus. The gross majority of the viruses (n = 39) was detected in a single mosquito species and only 10 viruses were detected in two or three different closely related mosquito species that belonged to the same mosquito genus indicating

a high host specificity (*Table 1*). Unfortunately, ordination techniques like non-metric multidimensional scaling did not converge due to too few positive findings of viruses in the pools (data not shown).

We often detected multiple viruses in one mosquito pool, especially in *Culex* pools, indicating possible co-infections mainly between GOLV, HEBV, CAVV, and FERV. These viruses were all associated with *C. nebulosus* as main mosquito host and consequently often found together in pooled mosquitoes of this species. We found significant positive associations between GOLV and HEBV, as well as between GOLV and CAVV (*Figure 6*). The association between GOLV and HEBV strongly increased (Spearman's rho = 0.84, p<0.05) when considering only the host mosquito *C. nebulosus*. Here, also correlations of both viruses with FERV and Spilikins virus increased (rho ~ 0.45, p<0.01; results not shown).

Whether these mixed infections result from multiple infected individuals or from co-infected single mosquitoes could not be analyzed as no homogenates of individual specimens were available. All four viruses could be isolated together in cell culture from multiple pools. The replication of all four viruses in co-infected cell cultures was confirmed by quantitative real-time PCR suggesting no general inhibitory effect and the possibility of simultaneous infection of a single mosquito.

The three jonviruses, JONV, Spilikins virus, and Mikado virus, were each also associated with specific mosquito species, namely with *C. decens*, *C. nebulosus,* and *C. annulioris*, respectively. Interestingly, their partial RdRp sequences showed a high degree of variation in the third codon positions that did not alter the translated protein sequences (called synonymous substitutions). For example, JONV and Mikado virus diverged by approximately 20% in their nucleotide (nt) sequences but only by 7% in their amino acid (aa) sequences. Similarly, Spilikins virus and JONV showed nt and aa divergences of 25% and 12–15%, respectively. Fixed effects likelihood (FEL) analysis of 31 detected jonvirus sequences found significant (p-value<0.05) evidence of negative selection for 88 out of 117 codons. These findings could point towards adaptation to a specific mosquito host under purifying selection. Similar strains with mainly variation in the third codon position were also observed for two Cimodo virus strains, HEBV and Cimo peribunyavirus III, two strains of Cimo peribunyavirus I, two strains of Anopheles flavivirus, as well as Cimo flavivirus I and Cimo flavivirus VII.

## Virus prevalence patterns along the anthropogenic disturbance gradient

According to current hypotheses in infectious disease ecology, ecological perturbation and changes in community composition are expected to influence transmission dynamics of infectious diseases (*Keesing et al., 2010*; *Randolph and Dobson, 2012*; *Ostfeld and Keesing, 2000*). We thus next analyzed the prevalence patterns of all detected viruses as well as abundance of their mosquito host species along the anthropogenic disturbance gradient.

Across the surveyed sites, mosquito group explained on average ~26% of the variation in viral association, followed by habitat (13%) and their statistical interaction (12%) (all p-values<0.001), 48% of the variance remained unexplained. The number of mosquitoes per pool did not have a strong additional effect (<1%, p>0.1). Principal coordinate analysis (PCoA) ordination plots show that the viral community primarily partitions by *C. nebulosus*, *Uranotaenia* sp., and *C. decens* (*Figure 6— figure supplement 1*), which can be related to their main habitats in primary forest, agriculture, and villages (*Figure 6—figure supplement 2*).

The highest virus richness was observed in the intermediately disturbed habitats secondary forest and agriculture (*Figure 7A*). However, the proportional number of tested mosquitoes was lower in the primary forest so that the richness in this habitat is likely underestimated.

The majority of the viruses (82%, n = 40) occurred with a low frequency of less than 10 positive samples in total (see *Table 1*). The cumulated MIR for all detected viruses was slightly higher in villages and at the camp sites compared to the other habitats (*Figure 7B*). This effect was mainly caused by the increasing prevalence of several bunyaviruses and one mesonivirus (*Figure 7—figure supplement 1A, B, F, and H*), while other taxa like reoviruses and iflaviruses were found in specific habitats (*Figure 7—figure supplement 1C and I*), or like flaviviruses and rhabdoviruses increased in prevalence towards pristine or intermediately disturbed habitats (*Figure 7—figure supplement 1E and G*). Of note, no amplification effect was observed in habitats with higher biodiversity (*Figure 7A and B*).

However, nine viruses were found more frequently with detection rates ranging from 12 to 42 strains (*Table 1*). As the sample set consisted of pooled mosquito specimens, we estimated viral infection rates

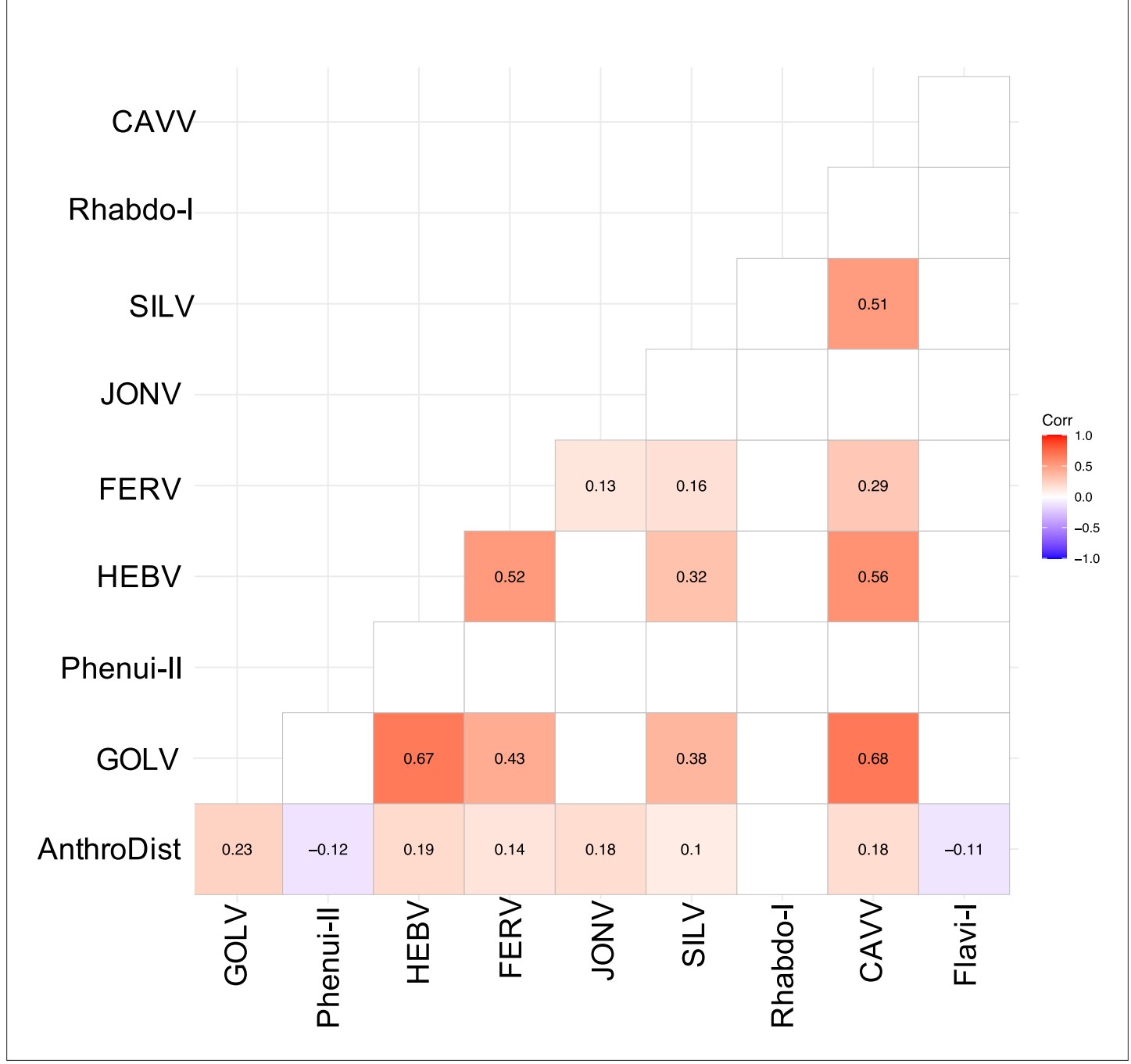

**Figure 6.** Spearman's rank correlation rho for the most abundant viruses. Only significant correlations are shown. AnthroDist refers to the gradient of anthropogenic disturbance from low to high. PF: primary forest; SF: secondary forest; A: agriculture; C: camp; V: village.

The online version of this article includes the following figure supplement(s) for figure 6:

**Figure supplement 1.** Ordination plot of the principal coordinate analysis (PCoA) showing that the viral community primarily partitions by *Culex nebulosus* (CulNeb), *Uranotaenia* species (Ura_spec), and *Culex decens* (CulDec).

**Figure supplement 2.** Ordination plot of the principle coordinate analysis (PCoA) showing that the viral community primarily partitions by *Culex nebulosus* (CulNeb), *Uranotaenia* species (Ura_spec) and *Culex decens* (CulDec) as shown in *Figure 6—figure supplement 1*, which can be related to their main habitats in primary forest, agriculture and villages.

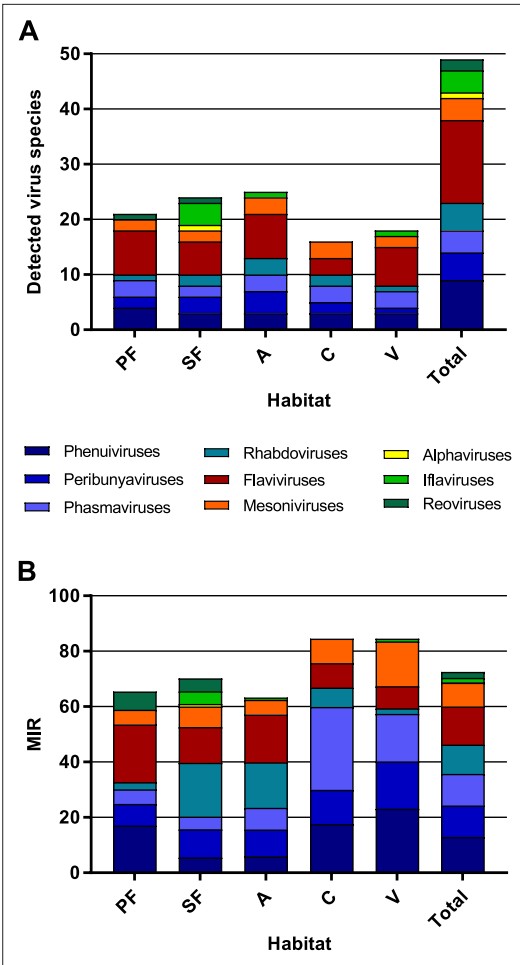

**Figure 7.** Richness and cumulative minimum infection rate (MIR) across all tested virus taxa. The number of distinct viruses (**A**) and the cumulated MIR per 1000 mosquitoes (**B**) were calculated for all habitat types and for the complete data set. Different virus taxa are shown in different colors. PF: primary forest; SF: secondary forest; A: agriculture; C: camp; V: village.

The online version of this article includes the following figure supplement(s) for figure 7:

**Figure supplement 1.** Cumulative minimum infection rate (MIR) per virus taxon.

in the different habitats using the two approaches MIR and MLE, which showed similar results (*Figure 8*). Four bunyaviruses (GOLV, HEBV, FERV, and Spilikins virus) and one mesonivirus (CAVV), all mainly associated with *C. nebulosus*, increased in prevalence towards disturbed habitat types (*Figure 8A–C and E*, left graphs, and *Figure 8H*). This is a trend that seemed to support the dilution effect hypothesis. In contrast, Cimo rhabdovirus I had its highest prevalence in the intermediately disturbed habitats (secondary forest and agricultural areas) (*Figure 8D*, left graph) and two viruses increased in prevalence towards the primary and secondary forest (Cimo phenuivirus II and Cimo flavivirus I) (*Figure 8F and I*). JONV prevalence slightly increased in the villages and more prominently at the camp sites compared to the other habitats (*Figure 8G*). The prevalence patterns corresponded for eight of the nine viruses to the relative abundance of the main mosquito host species (*Figure 8*). JONV was the only virus that showed no relationship between mosquito host abundance and virus prevalence, suggesting that other factors may influence JONV abundance. Cimo rhabdovirus I and JONV were both mainly associated with *C. decens* mosquitoes as host. However, in contrast to GOLV, HEBV, CAVV, and FERV, which were frequently detected together in *C. nebulosus* mosquitoes, Cimo rhabdovirus I and JONV were only twice found together in the same pool. This could hint to a possible interference between these viruses and might be a reason for the unusual prevalence pattern of JONV.

The probability of finding an infection with GOLV, HEBV, and CAVV was significantly higher in villages (p<0.1, *Supplementary file 2*). These viruses were mainly associated with *C. nebulosus* mosquitoes. Cimo phenuivirus II was associated with *Uranotaenia* sp. mosquitoes in the primary forest (p<0.1, *Supplementary file 2*), whereas JONV and FERV mainly occurred in the camps and villages, associated with *C. nebulosus*, *C. decens,* and other mosquito species. Cimo flavivirus I did not show a clear tendency, but mainly occurred in the primary and secondary forest as well as the agricultural sites in *Coquillettidia* sp. and other mosquito species. Cimo rhabdovirus I was almost exclusively associated with *C. decens* and the detection probability significantly increased in the secondary forest and the agricultural sites (p<0.1, *Supplementary file 2*).

We next investigated infection rates only in the main mosquito host species across the different habitat types. Only five of the nine viruses were detected frequently enough in their main mosquito host species (n > 10) to calculate host-specific infection rates. Surprisingly, no trend of increasing or decreasing virus prevalence was detected along the disturbance gradient (*Figure 8A–E*, right graphs) as would be expected in case of a dilution or amplification effect (*Ostfeld and Keesing, 2000*; *Levine et al., 2017*). Viral infection rates did not change considerably in their mosquito hosts between disturbed and undisturbed habitat types. Notably, the increase in prevalence of specific

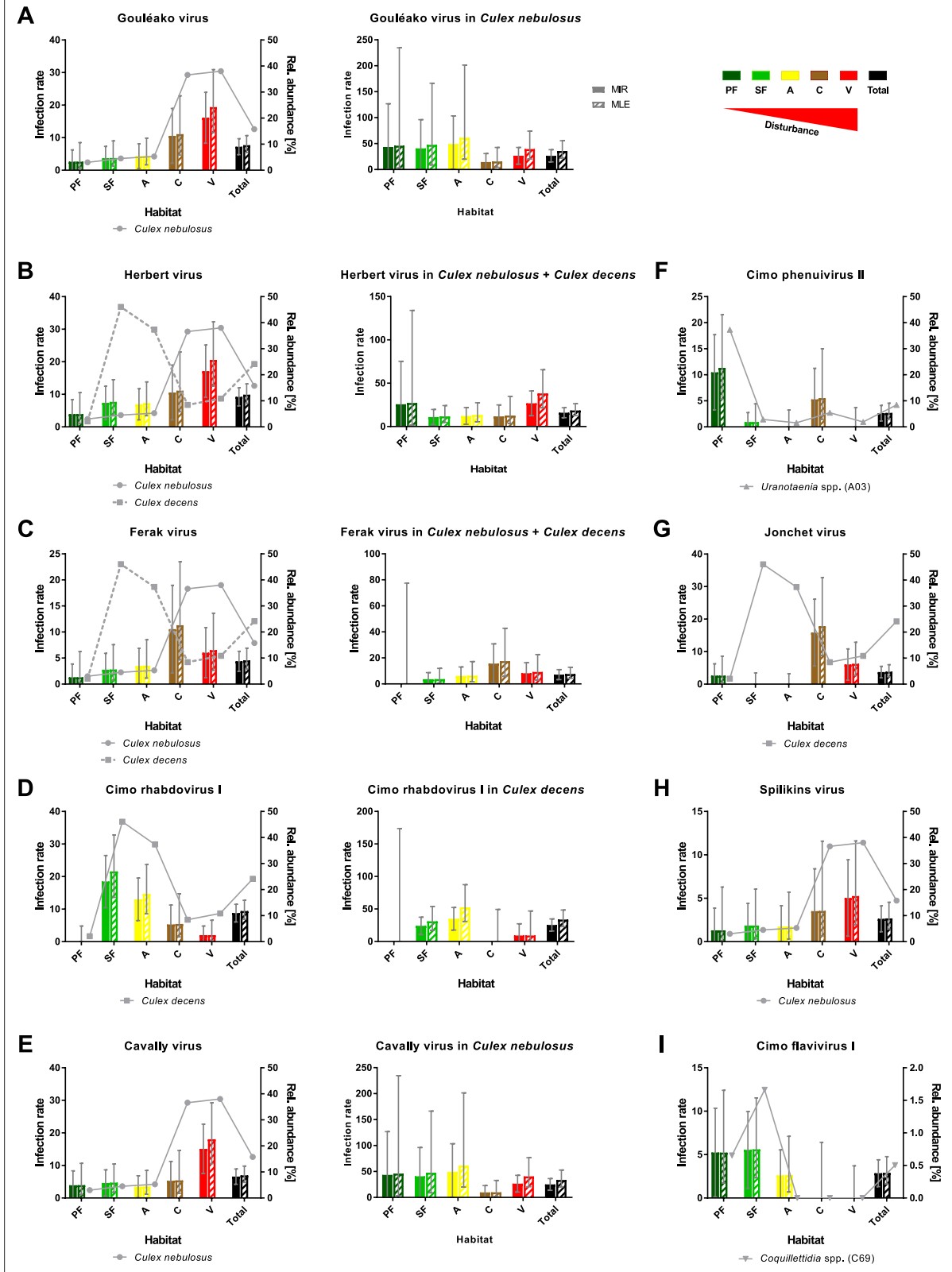

**Figure 8.** Prevalence patterns of selected viruses along the disturbance gradient. For all viruses that were detected in >10 pools, Gouléako virus (GOLV) (**A**), Herbert virus (HEBV) (**B**), Ferak virus (FERV) (**C**), Cimo rhabdovirus I (**D**), Cavally virus (CAVV) (**E**), Cimo phenuivirus II (**F**), Jonchet virus (**G**), Spilikins virus (**H**), and Cimo flavivirus I (**I**), the minimum infection rate (MIR) and maximum likelihood estimation (MLE) per 1000 mosquitoes of the whole data set were calculated for all habitat types (left graphs for **A–E**). The abundance of the main mosquito host species was plotted. The five viruses GOLV

*Figure 8 continued on next page*

*Figure 8 continued*

(**A**), HEBV (**B**), FERV (**C**), Cimo rhabdovirus I (**D**), and CAVV (**E**) occurred frequently enough in their main mosquito host species (>10 positive pools) to analyze their prevalence in these species. For these viruses, the MIR and MLE per 1000 mosquitoes of the respective species were calculated for all habitat types (right graphs). Significant differences in the infection probability with the most abundant viruses in the different habitats are shown in **Supplementary file 2**.

viruses resulted from shifts in the mosquito community composition along the gradient, which caused increased abundance rates of the main mosquito host species and concomitantly higher prevalence rates of the viruses they were carrying. We thus refer to this observation as abundance effect (summarized in **Figure 9**).

## Discussion

In this study, we analyzed the interplay between host community composition, habitat disturbance, and virus prevalence in a multi-host and multi-taxa approach. We characterized the genetic diversity of RNA viruses (multi-taxa) in an entire family of hosts (*Culicidae*; multi-host), which were sampled along an anthropogenic disturbance gradient. We subsequently studied prevalence patterns of all detected viruses per habitat type, as well as mosquito community composition per habitat type. We discovered an exceptionally high diversity of 49 distinct viruses, of which 34 were previously unknown members of seven different RNA virus families. We demonstrated that the majority of these viruses occurred at low minimum infection rates (MIRs) of 0.22–1.97 infected mosquitoes per 1000 tested mosquitoes. Nine viruses occurred more frequently across the disturbance gradient, of which five increased in prevalence from pristine to disturbed habitat types. We could show that the detection rates of these viruses corresponded to the abundance patterns of their specific mosquito host species. Importantly, the differences in virus prevalence were driven by the number of hosts present in a specific habitat type and not by changes in host infection rates. This interplay was named abundance effect. These data show that host community composition critically influences virus prevalence that has direct impact on our understanding of infectious disease emergence mechanisms.

The majority of the detected viruses grouped with ISVs in phylogenetic analyses, suggesting that vertebrates do not participate in their amplification and maintenance cycles. However, the novel viruses Cimo peribunyavirus I and II formed a monophyletic clade that shared a most recent common ancestor with viruses of the genera *Orthobunyavirus* and *Pacuvirus*. Orthobunyaviruses are arboviruses

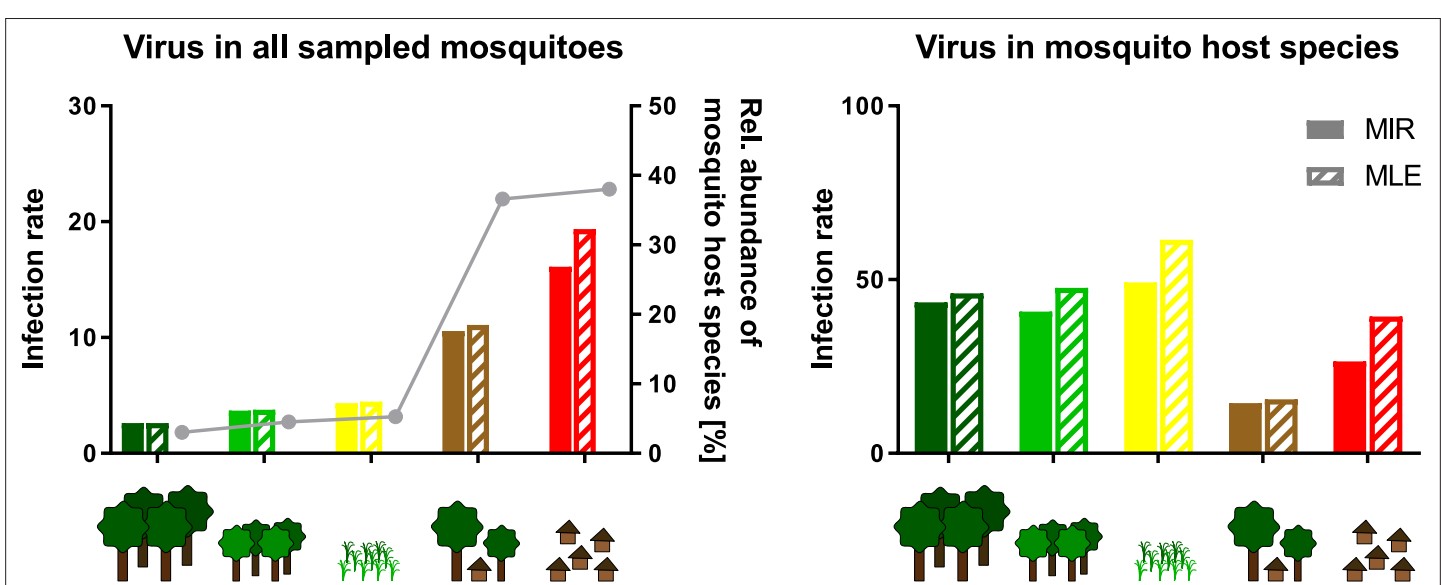

**Figure 9.** Schematic presentation of the abundance effect. Infection rates are shown for Gouléako virus (GOLV) in all sampled mosquitoes (representing minimum infection rate (MIR) and maximum likelihood estimation (MLE) values as shown in **Figure 8**) and only in the main mosquito host species, *Culex nebulosus*. The abundance of the main mosquito host species, *Culex nebulosus* is indicated by a gray line.

that infect a great variety of vertebrates including humans (*Hughes et al., 2020*). Pacuviruses were isolated from rodents and phlebotomine sandflies in Brazil (*Aitken et al., 1975*; *Rodrigues et al., 2014*). Thus, the amplification cycle of Cimo peribunyavirus I and II may involve vertebrates and may be more complex than that of the other detected viruses. Further research is necessary to assess the host range of Cimo peribunyavirus I and II. The ISVs detected here were previously unknown, and it is thus not known whether they are viral symbionts of mosquitoes or pathogenic viruses. Since these represent previously unknown viruses, no study investigating whether they induce symptoms of disease and/or influence life history traits of mosquitoes has been conducted. Either way, the study provides data for a better understanding of factors influencing the viral community composition of mosquitoes. Although the microbiome of invertebrates is widely unexplored, it has been shown that the microbiome of mosquitoes can have a negative effect on its vector competence (the ability of mosquitoes to transmit human- and livestock pathogenic viruses). In consequence, even if the viruses detected in this study would be symbionts of mosquitoes, they may affect the transmission efficiency of pathogenic viruses and affect spillover events of pathogenic arboviruses from sylvatic to urban cycles. Nevertheless, studying abundance patterns and geographic spread of ISVs in the light of habitat disturbance and shifts in host community composition can provide valuable insight into infectious disease dynamics, at least for directly transmitted viruses. Mosquito-borne viruses are among the major global health concerns, and the recent spread and epidemics caused by Zika and chikungunya viruses have exemplified that there is an urgent need to understand viral emergence processes (*Weaver et al., 2018a*). Using mosquito-specific viruses as model viruses has many advantages as these viruses are more frequently found than arboviruses and no vertebrate host is required for virus maintenance and amplification, which makes links between host community composition and virus abundance patterns easier to identify. Mosquitoes and their viruses are an ideal system to study such effects as this allows studying infectious disease dynamics in an entire family of hosts, which can be tested in high numbers.

Our observed virus prevalence patterns are in agreement with several studies stating a heterogeneous effect of biodiversity on pathogen prevalence or disease risk (*Randolph and Dobson, 2012*; *Salkeld et al., 2013*). Biodiversity can influence disease risk by different mechanisms like host regulation, changes in encounter rates or transmission rates, leading to an amplifying or diluting net effect (*Keesing et al., 2006*). Areas with a high host richness likely harbor a high pathogen richness that might act as a source pool of novel diseases upon habitat change and increased contact to humans (*Dunn et al., 2010*; *Keesing et al., 2010*). We observed the highest virus richness in intermediately disturbed and primary rainforest habitats supporting that biodiverse habitats are also rich in pathogens. However, highest viral prevalence rates were observed in villages and at camp sites, further supporting that intact or intermediately disturbed ecosystems also contain well-balanced spectra of viruses.

For those arboviruses that are transmitted by a generalist mosquito species and use a competent host profiting from disturbance, a dilution effect can occur at local scales (*Halliday et al., 2017*; *Ostfeld and Keesing, 2012*). Several studies observed a dilution effect for WNV with either increasing non-passerine or total bird diversity (*Allan et al., 2009*; *Ezenwa et al., 2006*; *Swaddle and Calos, 2008*) while others reported no protective effect of avian species richness on WNV prevalence (*Levine et al., 2017*; *Loss et al., 2009*). For tick-borne encephalitis virus, a dilution effect with increasing density of incompetent deer hosts was observed at local scale (*Cagnacci et al., 2012*). Likewise, the prevalence of the directly transmitted hantavirus sin nombre virus is reduced at sites with higher rodent diversity as the persistence of the main host species (deer mouse) is reduced at diverse sites (*Clay et al., 2009*). A similar pattern was observed in our sampling for the five viruses GOLV, HEBV, FERV, Spilikins virus, and CAVV belonging to four different families (*Phenuiviridae*, *Peribunyaviridae*, *Phasmaviridae*, and *Mesoniviridae*). All these viruses used mosquitoes of the species *C. nebulosus* as primary host, which seemed to profit from disturbance and increased in abundance in disturbed versus pristine habitats (38% vs. 3%).

The ecological mechanisms leading to changes in mosquito community composition due to anthropogenic disturbance are presently poorly understood. Environmental changes, including land use changes, but also changes in the host species composition have been suggested to act as drivers of vector-borne diseases, including important arboviral diseases like dengue (*Patz et al., 2000*; *Takken and Verhulst, 2013*; *Young et al., 2017*). Yet, very few studies have focused on *C. nebulosus*. It has

been reported as a sylvatic species in Nigeria, Benin, and Mozambique (*Abílio et al., 2020*; *Agwu et al., 2016*; *Lingenfelser et al., 2010*), but to the best of our knowledge this is the first study showing an increasing abundance of *C. nebulosus* in modified ecosystems. The possible reasons for the varying mosquito community compositions in the different habitat types might be the availability of suitable larval habitat sites and food sources, the abundance of predators, as well as differences in temperature and humidity (*Ferraguti et al., 2016*; *Roiz et al., 2015*; *Thongsripong et al., 2013*).

The prevalence of the five viruses associated with *C. nebulosus* was reduced in the diverse habitat types where the abundance of incompetent mosquito hosts increased and *C. nebulosus* was rarely found. It is striking that this effect was observed for five taxonomically different viruses infecting the same host species. This effect was not observed for closely related viruses using other mosquito species as host that did not increase in abundance in disturbed habitats. These data suggest that the adaptability of mosquito hosts to changed environments plays a more important role for the increase in prevalence of associated viruses than the phenotypic or genetic characteristics of these viruses. A similar observation was made by another study, which showed that the virome composition of three mosquito species captured along a rural to urban disturbance gradient in Thailand was defined primarily by the mosquito host species rather than the geographic location and that ISVs display a relatively narrow host range (*Thongsripong et al., 2021*).

Contrary effects can be caused by scale-dependent effects or depending on additive or substitutive community assemblies (*Johnson et al., 2015b*). Additionally, the most competent host can either increase or decline with disturbance, leading either to a dilution or amplification effect, respectively (*Halliday et al., 2017*). In our investigated sample set, two viruses (Cimo phenuivirus II, family *Phenuiviridae*, and Cimo flavivirus I, family *Flaviviridae*) with higher prevalence in pristine than in disturbed habitats were detected. As observed for the five viruses mentioned above (GOLV, HEBV, FERV, Spilikins virus, and CAVV), virus prevalence rates were closely linked to abundance rates of host mosquito species. Cimo phenuivirus II and Cimo flavivirus I were associated with *Uranotaenia* and *Coquillettidia* mosquitoes that declined in abundance in disturbed habitat types. An explanation of why such an effect is rarely found could be that model pathogens selected in the field of disease ecology are usually zoonotic pathogens that infect humans in disturbed habitats and are known to cause outbreaks. Thus, there may exist a bias towards those pathogens that thrive in disturbed habitat types. Pathogens that are rarely found in disturbed habitats may never be selected for disease ecology studies. Notably, the vast majority of detected virus species showed low infection rates without any evidence for effects of mosquito community changes on their abundance. These findings for 40 (out of 49) distinct viruses infecting different mosquito species suggest that host community disassembly does only influence a small fraction of the pathogen population. These examples underline why an unbiased multi-host and multi-taxa approach is crucial to uncover host community effects on pathogen prevalence patterns.

In other studies focusing on plant, vector-borne, indirectly and directly transmitted pathogens, the composition of the host community rather than total biodiversity was a predictor for pathogen distribution (*Randolph and Dobson, 2012*; *McLeish et al., 2017*). For example, (i) a low diversity of grassland plant species diversity was associated with a higher abundance of fungal plant pathogens (*Mitchell et al., 2002*), (ii) the disease and infection risk of populations of wild pepper (chiltepin) with different viruses increased with the level of human management, which was associated with decreased species diversity and host genetic diversity, and increased host plant density (*Pagán et al., 2012*), and (iii) the prevalence of four generalist aphid-vectored pathogens (barley and cereal yellow dwarf viruses) increased in grass host plants as local host richness declined, indicating that virus incidence in the focal host species is affected by the composition of the entire host community rather than by dependence on species diversity (*Mitchell et al., 2002*; *Pagán et al., 2012*; *Lacroix et al., 2014*). This is in agreement with the strong observed association between the prevalence of a certain virus and the abundance of its main mosquito host species in our study.

We detected novel strains of two previously characterized viruses, PCLV and Anopheles flavivirus, in *A. aegypti* and *Anopheles gambiae* mosquitoes, respectively, corresponding to previous findings of these viruses in these mosquito species in Asia and Africa (*Chandler et al., 2014*; *Fauver et al., 2016*; *Zhang et al., 2018*). This further supports the specific associations between certain viruses and mosquitoes on a broader geographical scale.

Mosquitoes harbor a diverse natural virome that can influence subsequent virus infections (*Hall et al., 2016*; *Junglen and Drosten, 2013*). We observed four frequently co-occurring viruses in *C.*

*nebulosus* mosquitoes (GOLV, HEBV, FERV, and CAVV) that also grew together in cell culture, while in contrast the two *C. decens*-associated viruses, Cimo rhabdovirus I and JONV, were rarely detected together. This could hint, on the one hand, at synergistic interactions and, on the other hand, at super-infection exclusion between different ISVs in mosquitoes. For several ISVs in the genera *Flavivirus* and *Alphavirus,* an interference with related arboviruses was observed (*Bolling et al., 2012*; *Nasar et al., 2015*; *Hobson-Peters et al., 2013*). A better knowledge of the virome of different mosquito species as well as studies on mutual interference might help to assess vector competence and possibilities of geographic spread.

Virus detection in mosquito homogenates independent of virus isolation extents potential virus findings but has the limitation that exogenous and integrated viruses can be discovered. The detection of likely integrated sequences derived from flavi- and rhabdoviruses in our mosquito sample is in agreement with frequent previous findings of NIRVS from these families in mosquito genomes (*Fort et al., 2012*; *Katzourakis et al., 2010*; *Lequime and Lambrechts, 2017*; *Whitfield et al., 2017*). NIRVS are more closely related to ISVs than to arboviruses, probably limiting their influence on vector competence. The transovarial transmission of ISVs might increase the chance of germline integrations (*Palatini et al., 2017*; *Olson and Bonizzoni, 2017*). NIRVS can be transcriptionally active and might play a role in immunity against related viruses by producing piRNAs (*Fort et al., 2012*; *Lequime and Lambrechts, 2017*; *Whitfield et al., 2017*; *Ter Horst et al., 2019*).

Collectively, our data show that only some viruses of a huge viral community benefited from ecosystem disturbance. This effect was found for five viruses (GOLV, HEBV, FERV, Spilikins virus, and CAVV) and was determined by abundance patterns of their mosquito hosts, which profited from habitat disturbance and strongly increased in numbers from pristine to disturbed habitats. In contrast, we also detected two viruses (Cimo phenuivirus II and Cimo flavivirus I) that were associated with mosquito hosts specific to primary habitat types. Detection rates of the associated viruses were also closely linked to host abundance, and these viruses were found with higher frequencies in the primary and secondary rainforest. However, our analyses have also shown that virus prevalence is not always a consequence of the abundance of host species in a certain area. The prevalence pattern of JONV did not follow the abundance pattern of *C. decens* mosquitoes and its abundance seems to be determined by unknown mechanisms, possibly by superinfection exclusion mediated by Cimo rhabdovirus I. The study of prevalence patterns of a broad genetic diversity of RNA viruses and their associated hosts allowed us to seek for general mechanisms influencing emergence and geographic spread of mosquito-associated viruses. No general dilution or amplification effect was observed. Instead, we identified changes in mosquito community composition causing an increase or decrease in the main host species as the most important factor determining virus prevalence patterns, an effect named abundance effect. Remarkably, host infection rates were not affected by higher host abundance in our study.

## Methods
### Mosquito collection

In total, 4562 female mosquitoes were collected in five habitat types along an anthropogenic disturbance gradient in the area of the Taï National Park in Côte d'Ivoire in 2004 (*Junglen et al., 2009b*). Habitat types represented pristine forest (PF), secondary forests (SF), agricultural areas (A), camps within primary forest (C)m and villages (V) in the order of low to high human disturbance (*Figure 1A*). Mosquitoes were collected using five CDC miniature light traps and one CDC gravid trap (developed by the U.S. Centers for Disease Control and Prevention, John W. Hock Company, USA) per sampling site, which were five sites in the PF, three sites in the SF, four sites in A (coffee, cacao, rice, and rice plantations), four camp sites (C), and two villages (V). Sampling was conducted for three consecutive days per sampling site from February to June 2004. The anthropogenic disturbance gradient was defined by apparent presence and density of humans as indicated by housing, for example, camps and villages. In addition, we accounted for the degree of plant species turnover from primary forest to agriculture. The camp is an exception as the vegetation is similar to the primary forest; however, we rated human presence higher and therefore grouped the camp closer to the village (*Figure 1A*). Mosquitoes were identified morphologically (*Junglen et al., 2009b*) and based on their COI sequences (*Folmer et al., 1994*). Mosquito heads were homogenized in 430 pools consisting

of 1–50 individuals according to species and sampling location (*Junglen et al., 2009b*), resulting in 98 pools (764 mosquitoes) from the primary forest, 98 pools (1083 mosquitoes) from the secondary forest, 100 pools (1153 mosquitoes) from agricultural areas, 66 pools (568 mosquitoes) from research camps, as well as 68 pools (994 mosquitoes) from two villages.

## RT-PCR screenings and sequencing

RNA was extracted from the pooled supernatants using the QIAamp Viral RNA Mini Kit (QIAGEN). The SuperScript III Reverse Transcriptase (Invitrogen–Thermo Fisher Scientific, Waltham, USA) was used for cDNA synthesis according to the manufacturer's instructions. Generic RT-PCR assays were established based on alignments of the RNA-dependent RNA polymerase (RdRp) sequences for the following taxa, peribunyaviruses, jonviruses, feraviruses, rhabdoviruses, flaviviruses, iflaviruses, orbiviruses, and Cimodo virus. The conventional nested RT-PCRs were designed based on sequence information from viruses, which have been previously isolated from these mosquitoes in cell culture (*Marklewitz et al., 2015*; *Zirkel et al., 2011*; *Hermanns et al., 2014*; *Junglen et al., 2009a*; *Junglen et al., 2017*; *Kallies et al., 2014*; *Marklewitz et al., 2011*; *Marklewitz et al., 2013*; *Quan et al., 2010*; *Zirkel et al., 2013*), and from all major arbovirus taxa. Primer sequences are listed in *Supplementary file 3*. In addition, we used previously described conventional RT-PCR assays for mesoniviruses (*Zirkel et al., 2013*), phenuiviruses (*Marklewitz et al., 2019*), flaviviruses (*Crochu et al., 2004*; *Moureau et al., 2007*), and alphaviruses (*Hermanns et al., 2017*) to test the samples. Due to the difficulties in the establishment of a generic RT-PCR assay for the unclassified taxon of negeviruses, it was not possible to include these viruses in the analyses. This entire approach allowed the detection of viruses regardless of isolation success in cell culture. PCR products were sequenced by Sanger sequencing (Microsynth AG, Balgach, Switzerland). In addition to the generic conventional RT-PCR assays, specific qPCRs were used to test for CAVV, Nsé virus, and Mikado virus (*Zirkel et al., 2013*). The entire RdRp motifs of the third conserved region were amplified from all sequences with more than 5% divergence to another sequence using primer walking. Selected virus isolates were sequenced by metagenomic sequencing as previously described (*Hermanns et al., 2017*).

## Genomic and phylogenetic analyses

All sequences were assembled and analyzed in Geneious R9.1.8 (*Kearse et al., 2012*). Viral sequences were categorized based on genetic similarity and phylogenetic analysis. Sequences were compared to the NCBI database using blastn and blastx. Sequences with <95% amino acid identity to known viruses were considered as putative novel virus species and named Cimo virus (acronym for *Côte d'I*voire and *mo*squito) with ascending numbering. Viruses isolated in cell culture and those with completely sequenced genomes received individual names. For phylogenetic analyses, the amino acid sequences (families *Phenuiviridae*, *Peribunyaviridae*, *Phasmaviridae*, *Rhabdoviridae,* and *Iflaviridae* and genus *Orbivirus*) or the nucleotide sequences (genera *Flavivirus* and *Alphavirus* and family *Mesoniviridae*) of the detected viruses and the related established virus species were aligned by MAFFT-E v7.308 (*Katoh and Standley, 2013*) in Geneious. An optimized maximum-likelihood phylogenetic tree with the substitution model based on Smart Model Selection (*Lefort et al., 2017*) as implemented in PhyML (mainly LG or GTR, respectively) and 1000 bootstrap replicates was calculated using PhyML (*Guindon et al., 2010*). For the detected jonviruses, the nonsynonymous and synonymous substitution rates were inferred using FEL as implemented in Datamonkey (*Weaver et al., 2018b*).

## Virus growth analyses

Viruses were isolated in cell culture as described before (*Junglen et al., 2009b*). For the four novel viruses, Sefomo virus, Mikado virus, Tafomo virus, and Sassandra virus, growth kinetics were performed using the mosquito cell line C6/36 (ECACC 89051705, *A. albopictus*; identity has been confirmed by NGS, tested negative for mycoplasma). C6/36 cells were cultivated and seeded as previously described (*Hermanns et al., 2020*). The cells were infected at an multiplicity of infection (MOI) of 0.1 and incubated for 3 d at 28, 30, 32, and 34°C to determine temperature sensitivity as previously described for dual-host and insect-specific bunyaviruses (*Marklewitz et al., 2015*). Every 24 hr, 75 µl supernatant was taken and RNA was extracted using the NucleoSpin RNA Virus kit (Macherey-Nagel). cDNA was synthesized using the SuperScript IV reverse transcriptase (Invitrogen–Thermo Fisher Scientific) according to the manufacturer's instructions. Specific quantitative RT-PCRs were established for

all four viruses. Respective primer and probe sequences are SefomoV-F: 5′-TGGTTGAGACCTTCTG AGACTTTTC-3′, SefomoV-R: 5′-CAAAGGCCATCCCGAAGTATC-3′, SefomoV-TM: 5′–6-FAM/CCAT-TACAC/ZEN/CTCATCCCTATTTCATGCTGG/Iowa Black-FQ-3′, MikadoV-F: 5′-GAGAACTGTCAA AAATGGAGAAGAGA-3′, MikadoV-R: 5′-AGATGGCACCATTTTCAGTGATATAG-3′, MikadoV-TM: 5′–6-FAM/GCCAACAGC/ZEN/CAATTAAGAGAATGA/Iowa Black-FQ-3′, TafomoV-F: 5′-AATCTGATCTGG AGGACGAGTTG-3′, TafomoV-R: 5′-GCTGTTGATTAGCTGTGCATGAT-3′, TafomoV-TM: 5′–6-FAM/ GGTTCTTGC/ZEN/TGGACCAGGTG/Iowa Black-FQ-3′, SassandraV-F: 5′-CATTTTGGAAGGAGAT TTTTCGA-3′, SassandraV-R: 5′-GATCAAATTTCCAATAGCCCATAAA-3′ and SassandraV-TM: 5′–6-FAM/ TGGACCTCA/ZEN/AGCGGATTCAACCGT/Iowa Black-FQ-3′.

## Statistical analysis

### Virus prevalence

To estimate virus prevalence, the IR and MLE per 1000 mosquitoes were calculated using the Excel Add-In PooledInfRate, version 4.0 (**Biggerstaff, 2009**). Virus data analyses were performed in GraphPad Prism 7.04 (GraphPad Software, San Diego, USA).

### Biodiversity analyses

All biodiversity analyses were done with R version 3.6.2 (**R Development core Team, 2019**). We used rarefaction curves and Hill numbers of diversity order q = 0 (species richness) to compare mosquito communities between the five habitats **Chao et al., 2014**; R-package 'iNEXT' (**Hsieh et al., 2016**; **Chao and Hsieh TCM, 2020**). To this end, we summed the number of mosquito species per habitat and excluded non-defined species from the analysis. For detection of underlying gradients, we used the Bray–Curtis dissimilarity index **Legendre and De Cáceres, 2013**; R-package 'vegan' (**Oksanen et al., 2019**) and hierarchical cluster analysis to group main mosquito species according to their habitat association (R-package 'pheatmap'). To assess whether there are significant associations between these mosquito groups and habitat, we fitted a loglinear model (generalized linear model with 'log' link and Poisson error distribution) to the number of mosquitoes counted per group as response variable, with mosquito group and habitat as covariates in interaction. Since there were no counts of *Coquillettidia* species in camp and village, we added a small constant of 1 to all responses to avoid singularities in the model outcome and to be able to model it with a log transformation. We assessed significance of the single terms using a log likelihood ratio test (LRT) of this saturated model.

We used Spearman's rho on the full data set to assess which viruses were associated with each other in the different pools. We focused our analyses on abundant viruses we had found at least 10 times in all pools. For assessing virus–habitat relationships expressed as the probability of detecting one of the abundant viruses in a specific habitat, we fitted a generalized linear mixed model with binomial error structure and logit link and the respective virus type detected (**Swei et al., 2020**) or not (0) as response. As covariates, we included the number of mosquitoes per pool and the habitat. We included the number of mosquitoes per pool as controlling variable because the probability to detect any virus in a pool logically increases when more potential hosts are combined into a pool. Pairwise differences in virus infection probabilities in the habitats were assessed post hoc with Tukey contrasts (R-package multcomp).

In order to determine the relative contributions and amount of variance explained by host and habitat in partitioning the viral community, we used permutational multivariate analyses of variance (PERMANOVA) tests based on Jaccard distance (presence or absence of virus in the pool) (package 'vegan,' function adonis2). We entered mosquito host taxonomy in interaction with habitat and the number of mosquitoes per pool; we only used the main mosquito groups as well as pools with at least one presence of a viral OTU into the analysis, thereby reducing the data set to 122 entries.

## Nucleotide sequence accession numbers

The viral sequence fragments and genomes as well as the potential NIRVS were assigned the GenBank accession numbers MZ202249-MZ202305 and MZ399697-MZ399709, respectively. Accession numbers of representative sequences are listed in *Table 1*.

## Acknowledgements

We thank the Ivorian Ministry of Environment and Forest, the Ministry of Research, and the directorship of the Taï National Park for the opportunity to conduct this research. We thank all field assistants for help in the field, the Taï chimpanzee project for logistic support during field work, and Fabian Leendertz for helpful discussions regarding design of fieldwork. We are grateful to Verena Heyde, Christian Hieke, and Friederike Schröder for excellent laboratory assistance. The work was funded by the Federal Ministry of Education and Research (grant agreement number 01KI1716) as part of the Research Network Zoonotic Infectious Diseases and by the Deutsche Forschungsgemeinschaft (under grant agreement numbers JU2857/3-2 and DR772/10-2).

## Additional information

### Funding

| Funder | Grant reference number | Author |
|---|---|---|
| Federal Ministry of Education and Research | 01KI1716 | Sandra Junglen |
| German Research Foundation | JU2857/3-2 | Sandra Junglen |
| German Research Foundation | DR772/10-2 | Sandra Junglen |

The funders had no role in study design, data collection and interpretation, or the decision to submit the work for publication.

### Author contributions

Kyra Hermanns, Conceptualization, Data curation, Formal analysis, Investigation, Visualization, Writing - original draft, Writing – review and editing; Marco Marklewitz, Florian Zirkel, Anne Kopp, Investigation, Writing – review and editing; Stephanie Kramer-Schadt, Formal analysis, Visualization, Writing – review and editing; Sandra Junglen, Conceptualization, Resources, Supervision, Funding acquisition, Investigation, Writing - original draft, Writing – review and editing

### Author ORCIDs

Stephanie Kramer-Schadt http://orcid.org/0000-0002-9269-4446
Sandra Junglen http://orcid.org/0000-0002-3799-6011

### Decision letter and Author response

Decision letter https://doi.org/10.7554/eLife.66550.sa1
Author response https://doi.org/10.7554/eLife.66550.sa2

## Additional files

### Supplementary files

• Supplementary file 1. Model estimates (base: habitat primary forest and mosquito group *Anopheles* sp.) for the counts of mosquito individuals per mosquito group and habitat. Significant predictors show the difference to the baseline combination. CulAnn: *Culex annuloris*; CulDec: *Culex decens;* CulNeb: *Culex nebulosus;* Cul_spec: other *Culex* species; Ura_spec: *Uranotaenia* species; Coq_spec: *Coquillettidia* species; Ano_spec: *Anopheles* species; others: all other grouped species. PF: primary forest; SF: secondary forest; A: agriculture; C: camp; V: village.

• Supplementary file 2. Significant differences in the infection probability with the most abundant viruses in the different habitats (only Tukey contrasts are shown; bold: p<0.05; italic: p<0.1). No significant differences were detected for FERV (n = 20), JONV (n = 17), Spilikins virus (n = 12), and Cimo flavivirus I (n = 13; no findings in camp and village). Combinations with zero virus detections shown in light gray. All virus detections were positively associated with the number of mosquitoes per pool (model estimates not shown).

• Supplementary file 3. Primer pairs used for generic RT-PCR assays.

• MDAR checklist

### Data availability

The viral sequence fragments and genomes as well as the potential non-retroviral integrated RNA virus sequences (NIRVS) were assigned the GenBank accession numbers MZ202249–MZ202305 and MZ399697–MZ399709, respectively.

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
