## [Editor Report]

This paper explores the drivers of viral and host composition in natural and disturbed ecosystems. The authors make an important contribution to knowledge on the diversity of mosquito-specific viruses, describing the genetic diversity of RNA viruses from the family *Culicidae*. The paper will be of interest to scientists in the fields of virology, entomology, ecology and epidemiology. The data are of high quality and have been rigorously assessed.

---

## [Decision Letter]

**Decision letter after peer review:**

Thank you for submitting your article "Mosquito community composition shapes virus prevalence patterns along anthropogenic disturbance gradients" for consideration by *eLife*. Your article has been reviewed by 3 peer reviewers, and the evaluation has been overseen by a Reviewing Editor and George Perry as the Senior Editor. The following individuals involved in review of your submission have agreed to reveal their identity: Camila Gonzalez (Reviewer #2); Gabriel Hamer (Reviewer #3).

Essential revisions:

1. There is a large amount of data that has undergone very comprehensive analysis but the manuscript needs to be re-written substantially in the context of what is currently known about the transmission and ecology of insect-specific viruses. It appears premature to test hypotheses on ecological patterns of human/vertebrate diseases with these data on mosquito specific viruses.

2. The paper analyzes data on mosquito specific viruses under the lens of pathogens with public health importance, which is confusing. Clarification is needed as to why analyzing insect-specific virus prevalence and diversity may or may not serve as a model for the study of typical arboviruses, considering the differences in their maintenance in nature.

3. A caveat of the study is that the (likely) mechanisms of transmission of the viruses identified in the paper were not discussed. Mosquito specific viruses can be transmitted vertically or horizontally, and are in general strongly associated with the environment, but not related with other hosts such as vertebrates. From this perspective, the ecology of transmission of these viruses should not be compared to pathogens that use vertebrate hosts.

4. The authors found 49 viruses, but emphasize the ecological relevance of their findings to five viruses with increased prevalence from pristine to disturbed habitats, to show a dilution effect. This should be discussed.

5. It appears that nearly half of the mosquitoes in three of the study sites could not be identified to species. This appears problematic for the estimation of host (mosquito) richness and diversity along the anthropogenic gradient.

6. Viral taxonomy is also complicated and this study is presenting many new viruses which, based on partial or whole sequencing, are putative novel viruses. For many of the putative new viruses, only small sequences of less than 1200 nt were analysed. Granted that the RdRp is the most conserved gene, how was the 5% demarcation for a new species determined when established criteria differ to this. It is not clear how many of these novel viruses would be accepted by current practices endorsed by the International Committee on Taxonomy of Viruses. The viral taxa uncertainty add complexity for the current analysis. How many of these viral lineages that cluster together are variants of the same virus? How many are unique taxonomic units? This has important consequences on the application of these data to the analyses conducted in this study.

7. Many of these viruses the authors documented are mostly Insect-specific viruses (ISVs). But it also appears that several could be amplified by vertebrate hosts with poorly understood natural history and for the purposes of this study, all of the viral taxa appear to be grouped together. The inclusion of all viruses is therefore somewhat confounding given the very different natural history associated with these viruses. You frequently refer to 'hosts' throughout the manuscript and for ISVs, the host would likely only be mosquitoes but for arboviruses involving vertebrate amplification hosts, the hosts would be both the mosquitoes and the vertebrates. This study did not quantify any aspect of vertebrate host abundance, diversity, or richness across the gradient. Since most of this study focuses just on the ISVs as a unique system to test the hypotheses, it would be interesting if the authors restricted the analysis to just those viruses with higher probability of being restricted to mosquitoes (e.g. based on phylogenetic placement) to see if the results remain the same.

8. You report an anthropogenic disturbance gradient from primary forest to village habitat but how was this quantified? How is a village more disturbed than an agricultural field (rice plantation?)? The method to rank these study sites, which becomes important for the analysis, was not described in the methods.

9. Also, along this topic of study sites, it appears you really only had one replicate of each of the study site type. To test these hypotheses on how host communities influence viral communities it would seem prudent to have had multiple replicates of each study area.

10. A suggested important contribution of the paper is the finding of an "abundance effect", suggesting that higher prevalence in degraded ecosystems is the result of host abundance, but additional ecological information is missing on the potential mechanisms leading to this effect. Breeding sites may be a main source of variation in species composition and abundances among habitats, but no comments on this are found on the manuscript.

11. Some additional useful information could be provided to better understand mosquito sampling, for instance: the number of traps used, duration of sampling in each locality, and sampling dates to understand if there could be seasonal variation.

12. Mesoniviruses: Some mesonivirus species have been shown to be expectorated by mosquitoes, which explains their presence in multiple mosquito genera and suggests multiple modes of transmission. Perhaps insect-specific bunyaviruses could be the same given their detection in multiple mosquito genera?

13. Please clarify whether the numbers of pools of each mosquito species collected at each site are the same as in Table 1 of ref 29. I wanted to be able to see the actual numbers of mosquitoes collected at each location.

14. In line 97, it is stated that the aim was to capture all major taxa of arthropod-associated viruses, but I note that no assays were included to detect negeviruses, which are a growing taxon of insect-specific viruses.

15. The Results section is too long, particularly with detailed virologic information, and includes parts that should go on the discussion.

I strongly suggest to include SI Figure 1 (Biodiversity analyses) as a figure in the text since the main conclusions are based on variation of mosquito communities´ composition, and this is the result showing it.

16. Ln. 128: The authors state "primer and sequences and cycling conditions are available upon request". If this is the first study to present the results from these newly designed primers, they should be included in this publication. Perhaps as supplemental material. In fact right not it is hard for me to know if you had 8 primer pairs for these RT-PCRs or if any targets overlapped and shared a forward sequence for example. In the following sentences you say you used previously published assays as well, which appear to be a combination of conventional and real-time assays, so now it is getting hard to untangle which results came from which PCR.

*Reviewer #1 (Recommendations for the authors):*

The authors have done an excellent job of displaying very complex data, but I do believe that the methods for the transmission of these viruses in nature needs to be taken into consideration. Granted though, that there are likely to be differing scenarios for different viral families. E.g. Flaviviruses: Numerous groups have shown vertical transmission as the mechanism for persistence of insect-specific flaviviruses in nature. Furthermore most insect-specific flavivirus species have been detected in only one species of mosquito suggesting strong evolution of the virus with their mosquito host.

Mesoniviruses: Some mesonivirus species have been shown to be expectorated by mosquitoes, which explains their presence in multiple mosquito genera and suggests multiple modes of transmission. Perhaps insect-specific bunyaviruses could be the same given their detection in multiple mosquito genera?

Please clarify whether the numbers of pools of each mosquito species collected at each site are the same as in Table 1 of ref 29. I wanted to be able to see the actual numbers of mosquitoes collected at each location.

In line 97, it is stated that the aim was to capture all major taxa of arthropod-associated viruses, but I note that no assays were included to detect negeviruses, which are a growing taxon of insect-specific viruses.

For many of the new viruses presented in this study, that have not been published previously, relatively small sequences of 500 – 1200 nt have been used to assign viruses as new species. In terms of the flaviviruses, it is generally considered that a new virus species must have >16% sequence identity (nt) to be considered a distinct species.

Please clarify that in Figure 8, that the left hand graph is all detections of the virus, despite the mosquito species – this didn't come across clearly in the figure legend.

Were there any attempts to obtain isolates of Cimo peribunyaviruses I and II?

*Reviewer #2 (Recommendations for the authors):*

The Results section is too long, particularly with detailed virologic information, and includes parts that should go on the discussion.

I strongly suggest to include SI Figure 1 (Biodiversity analyses) as a figure in the text since the main conclusions are based on variation of mosquito communities´ composition, and this is the result showing it.

*Reviewer #3 (Recommendations for the authors):*

Ln. 21: Consider change to "…on the mechanism of emergence."

Ln. 22: Consider change to "...often observe contradictory results."

Ln. 27: "was" should be "were".

Ln. 28: You reference congruence to the dilution effect hypothesis here. But then the final statements of the discussion say 'No general dilution or amplification effect was observed'. Given the comments above, I have trouble seeing evidence to support conclusions on this topic.

Ln. 128: The authors state "primer and sequences and cycling conditions are available upon request". If this is the first study to present the results from these newly designed primers, they should be included in this publication. Perhaps as supplemental material. In fact right not it is hard for me to know if you had 8 primer pairs for these RT-PCRs or if any targets overlapped and shared a forward sequence for example. In the following sentences you say you used previously published assays as well, which appear to be a combination of conventional and real-time assays, so now it is getting hard to untangle which results came from which PCR.

Table 1. Would be good if the table legend or footnotes could explain the abbreviations in the column headers. I think these refer to study sites. Is there a way to sum these study sites and present a total number of unique viruses from each of the study sites? Would this be the number displayed in Figure 7A?

Ln. 260. You are using the term 'insect-restricted' here and in a couple other locations. Is this the same as 'insect-specific'? If so, I suggest staying consistent with insect-specific which you already use elsewhere in the manuscript. Or you could say these viruses are restricted to mosquitoes if that is appropriate in context.

Figure 2. For all these phylogenetic trees, it is hard to see how large of a gene region (or entire gene?) was used to build these. Would be helpful to include the number of base pairs, either in legend or in methods section.

Ln. 356. I'm not familiar with this process to attempt to identify if a virus is insect-specific by increasing temperature using insect cells (C6/36). But this sounds strange because holding C6/36 at 34C doesn't sound good for the cells so did the virus not replicate because the cells were all dead? Do you have a control of some kind showing C6/36 are still healthy at 34C? Did you attempt to grow these viruses in vertebrate cells?

Ln. 365. You are presenting the detection of viral integration into the mosquito genome but it is not clear how these observations were treated in the context of the analysis of viral taxonomy across the study sites. I assume there were ignored.

Ln. 394. Is it better to say "…pools were found positive at least one virus"?

Figure 7: Once again, would be good to identify what the habitat abbreviations refer to in legend.

Ln. 560. Chikungunya should be lower cased.

Ln. 609-611. True, so you should pull in the relevant literature from plant viruses (and in particular grasses) which have been used as a system to study the relationship between viral prevalence and host community composition.

---

## [Author Response]

Essential revisions:1. There is a large amount of data that has undergone very comprehensive analysis but the manuscript needs to be re-written substantially in the context of what is currently known about the transmission and ecology of insect-specific viruses. It appears premature to test hypotheses on ecological patterns of human/vertebrate diseases with these data on mosquito specific viruses.

Thank you very much for carefully reading the manuscript and your comment. With this study we aimed to get insight into fundamental questions in the field of disease ecology. Our main aim was to explore how a host community structure, such as number of species, relative abundance of species, and relative number of individuals per species in a community, influences the abundance, prevalence and spread of pathogens using mosquitoes and their associated viruses as a multi-host and multi-pathogen model system. We studied the dynamics of mosquito-associated viruses in relation to ecosystem and mosquito community changes. We revised the manuscript to clarify that we are not testing hypotheses on ecological patterns of human and vertebrate diseases. However, our study provides important insight how the composition of mosquito communities can affect the abundance and prevalence of mosquito-associated viruses. In consequence, these data may help to gain further insight into the dynamics of mosquito-borne viruses with respect to human and vertebrate diseases tackled from the vector part of the transmission cycle.

Mosquitoes and their viruses are an excellent system to study the effect of host community composition on pathogen abundance and prevalence patterns as mosquito communities can easily be sampled in high numbers and show high infection rates with mosquito-specific viruses. In contrast, mosquito-associated viruses causing disease in humans and vertebrates are rarely found in pristine habitats in nature and it is much more difficult to assess how ecosystem and host community changes affect their abundance. In the light of large-scale land use changes and unprecedented biodiversity loss, it is of utmost importance to understand how these ecological changes affect mosquito species composition and abundance of their associated pathogens. The data generated in this study will thus help to elucidate emergence processes of mosquito-associated viruses at the interface of undisturbed tropical ecosystems and anthropogenic landscapes.

We added the following paragraph to the introduction summarizing what is currently known about the ecology and transmission of insect-specific viruses (lines 88 – 106):

“In addition to the vertebrate-pathogenic arthropod-borne viruses (arboviruses), e.g. dengue virus (DENV), yellow fever virus (YFV) and Zika virus (ZIKV) (33, 34), mosquitoes can also be infected with so called insect-specific viruses (ISVs), which cannot infect vertebrates due to several host restriction barriers, e.g. sensitivity to higher (body) temperatures and inability to replicate in vertebrate cells (35, 36). All arbovirus taxa are embedded in a much greater phylogenetic diversity of ISVs, which suggests that arboviruses evolved from ancestral arthropod-restricted viruses (35-38). The mode of transmission of most ISVs in nature remains largely unknown but is suggested to rely on direct transmission. For some insect-specific flaviviruses, vertical transmission to the progeny was described and is considered to play an important role for virus maintenance in nature (35, 39-41). Additional transmission mechanisms of ISVs may include shared food sources, ectoparasites or venereal transmission (reviewed in (37, 42, 43)). It has been shown that at least some flavi-, mesoni and bunyaviruses are expectorated during feeding which may provide a route for horizontal virus transmission (44-47). However, transmission dynamics may differ among virus species and viral families and data are lacking describing maintenance and transmission of ISVs related to arboviruses (43). Knowledge on the natural transmission of ISVs has been mainly studied for entomopoxviruses and baculoviruses which infect other insects than mosquitoes (48). Due to their high abundance in natural mosquito populations, ISVs can thus serve as model systems to study how changes in mosquito species assembly affect associated virus abundance.”

2. The paper analyzes data on mosquito specific viruses under the lens of pathogens with public health importance, which is confusing. Clarification is needed as to why analyzing insect-specific virus prevalence and diversity may or may not serve as a model for the study of typical arboviruses, considering the differences in their maintenance in nature.

Many thanks for this comment. We are sorry if it was not clear that we are not focussing on pathogens with public health importance. However, we are studying how the structure of a host community affects infection rates. Such data may provide general insight into linkages between a host community structure and infectious diseases including diseases affecting humans. Infectious diseases are transmitted among individuals of a community, and it is so far not clear how a host community structure influences transmission dynamics of infectious diseases. As outlined above (see response 1), in this study we are testing how the structure of a natural host community influences pathogen abundance and infection rates using viruses that naturally infect mosquitoes in a natural system. Importantly, this study system is a multi-host and multi-pathogen system that allows to assess host-pathogen-specific differences depending on host and pathogen traits, as well as to identify common patterns among different host and pathogen species. So far, studies on multi-host and multi-pathogens have rarely been conducted and if so, were mostly based on a theoretical framework. Empirical data from natural communities, particularly empirical data analysing effects of a mosquito community structure on viral infections, are hardly available. Notably, in light of the current unprecedented biodiversity loss it is unclear how ecological changes such as deforestation and agricultural intensification affect the mosquito community composition, how the mosquito community is structured in changed landscapes and in turn how this affects viruses that are carried by mosquitoes that are resilient to disturbance.

We carefully revised the manuscript to make this clear. The respective paragraph in the introduction was revised and now reads (lines 82-106):

“Here, we provide a comprehensive analysis combining fields of community ecology and virology to understand how viral richness depends on host richness, and which role host-habitat associations play for viral abundance patterns using viruses that naturally infect mosquitoes in a natural system. Habitat alterations such as agricultural development and urbanization promote thriving of disease-transmitting mosquito species (25-29), but it is less clear how such changes in mosquito species disassembly affect the abundance and prevalence of viruses. From the limited amount of available studies, these mostly focused on effects of land use change on one specific virus (30-32). In addition to the vertebrate-pathogenic arthropod-borne viruses (arboviruses), e.g. dengue virus (DENV), yellow fever virus (YFV) and Zika virus (ZIKV) (33, 34), mosquitoes can also be infected with so called insect-specific viruses (ISVs), which cannot infect vertebrates due to several host restriction barriers, e.g. sensitivity to higher (body) temperatures and inability to replicate in vertebrate cells (35, 36). All arbovirus taxa are embedded in a much greater phylogenetic diversity of ISVs, which suggests that arboviruses evolved from ancestral arthropod-restricted viruses (35-38). The mode of transmission of most ISVs in nature remains largely unknown but is suggested to rely on direct transmission. For some insect-specific flaviviruses, vertical transmission to the progeny was described and is considered to play an important role for virus maintenance in nature (35, 39-41). Additional transmission mechanisms of ISVs may include shared food sources, ectoparasites or venereal transmission (reviewed in (37, 42, 43)). It has been shown that at least some flavi-, mesoni and bunyaviruses are expectorated during feeding which may provide a route for horizontal virus transmission (44-47). However, transmission dynamics may differ among virus species and viral families and data are lacking describing maintenance and transmission of ISVs related to arboviruses (43). Knowledge on the natural transmission of ISVs has been mainly studied for entomopoxviruses and baculoviruses which infect other insects than mosquitoes (48). Due to their high abundance in natural mosquito populations, ISVs can thus serve as model systems to study how changes in mosquito species assembly affect associated virus abundance.”

3. A caveat of the study is that the (likely) mechanisms of transmission of the viruses identified in the paper were not discussed. Mosquito specific viruses can be transmitted vertically or horizontally, and are in general strongly associated with the environment, but not related with other hosts such as vertebrates. From this perspective, the ecology of transmission of these viruses should not be compared to pathogens that use vertebrate hosts.

The reviewer is right that the transmission dynamics differ between ISVs and arboviruses. The mechanism of transmission for the viruses identified in this study is not known. Nearly all of the here identified viruses are most likely insect-specific and vertebrates are most likely not involved in their transmission. The text has been revised accordingly focussing on ISVs (also see responses 1 and 2).

4. The authors found 49 viruses, but emphasize the ecological relevance of their findings to five viruses with increased prevalence from pristine to disturbed habitats, to show a dilution effect. This should be discussed.

Thank you for pointing this out. Please note that abundance patterns were analysed and discussed for all 49 viruses (results lines 156 – 234 and discussion lines 359 – 472). However, since only nine of these viruses were detected frequently, and were found to differ in their abundance among the sampling sides, further analyses regarding ecological hypotheses explaining these abundance patterns were thus limited to these nine viruses. For five of these, the detection rates increased in disturbed habitat types, and we found that this effect could be explained by a turnover in the mosquito species composition and a higher abundance of a single mosquito species, *Culex nebulosus*, which is the host for all of the five frequently detected viruses that increased in abundance in disturbed habitat types.

Regarding the other 40 viruses, we added the following text to the Discussion section (discussion lines 442 – 447):

“Notably, the vast majority of detected virus species showed low infection rates without any evidence for effects of mosquito community changes on their abundance. These findings for 40 distinct viruses infecting different mosquito species suggest that host community disassembly does only influence a small fraction of the pathogen population. These examples underline why an unbiased multi-host and multi-pathogen approach is crucial to uncover host community effects on pathogen prevalence patterns.”

5. It appears that nearly half of the mosquitoes in three of the study sites could not be identified to species. This appears problematic for the estimation of host (mosquito) richness and diversity along the anthropogenic gradient.

It is true that this is a drawback of the analyses, however, basically affecting the species richness curves. Here, although we observed the tendency that richness of PF (primary forest) > C (camp sites) > SF (secondary forest) and A (agriculture) > V (villages), the rarefaction curves show that we only get a significant difference (non-overlapping CI) between PF (C) and V, i.e. the two extremes, with a gradient in between. The proportion of missing mosquito identification actually follows this trend, with the proportion of non-identification being highest in primary forest and lowest in villages (PF 40% , SF 40% , A 24% , C 26% , V 14% ). That means, the chances of higher species richness in primary forest is most likely even larger compared to the small proportion of non-identifications in villages, with a similar gradient in between. These results undermine the general tendency we report in the manuscript.

This caveat of our study is now addressed in the manuscript (lines 130 – 134):

“A large fraction of 40%, 40% and 26% of the sampled mosquitoes in the primary and secondary forest as well as at the camp sites, respectively, could not be identified to species level either due to morphological damage or to limitations of taxonomic keys. The actual number of different species in these habitats is therefore likely to be higher and the relative abundance of some mosquito species may have been underestimated.”

6. Viral taxonomy is also complicated and this study is presenting many new viruses which, based on partial or whole sequencing, are putative novel viruses. For many of the putative new viruses, only small sequences of less than 1200 nt were analysed. Granted that the RdRp is the most conserved gene, how was the 5% demarcation for a new species determined when established criteria differ to this. It is not clear how many of these novel viruses would be accepted by current practices endorsed by the International Committee on Taxonomy of Viruses. The viral taxa uncertainty add complexity for the current analysis. How many of these viral lineages that cluster together are variants of the same virus? How many are unique taxonomic units? This has important consequences on the application of these data to the analyses conducted in this study.

Thank you for pointing this out and we are sorry if this was not well explained before. Different virus species are generally defined by at least 5% genetic distance within their protein sequences of a specific gene (in most cases the RdRp protein or a combination of different genes) according to the species demarcation criteria of the International Taxonomy of Viruses (ICTV). The viral sequence fragments generated in this study originate from the highly conserved core region of the RdRp gene. As this region is the most conserved part of the RdRp, a 5% threshold based on this highly conserved region can thus be considered as a highly conservative threshold as the genetic distance between two sequences will increase when taking into account less conserved genome regions of the RdRp gene. We thus believe that all 49 viral sequences detected in this study are unique taxonomic units belonging to different species. All analyses presented in this study are based on this assumption.

7. Many of these viruses the authors documented are mostly Insect-specific viruses (ISVs). But it also appears that several could be amplified by vertebrate hosts with poorly understood natural history and for the purposes of this study, all of the viral taxa appear to be grouped together. The inclusion of all viruses is therefore somewhat confounding given the very different natural history associated with these viruses. You frequently refer to 'hosts' throughout the manuscript and for ISVs, the host would likely only be mosquitoes but for arboviruses involving vertebrate amplification hosts, the hosts would be both the mosquitoes and the vertebrates. This study did not quantify any aspect of vertebrate host abundance, diversity, or richness across the gradient. Since most of this study focuses just on the ISVs as a unique system to test the hypotheses, it would be interesting if the authors restricted the analysis to just those viruses with higher probability of being restricted to mosquitoes (e.g. based on phylogenetic placement) to see if the results remain the same.

Thank you for pointing this out. The analyses were indeed performed with viruses that are most likely restricted to mosquitoes. In total, 45 of the 49 viruses detected in this study grouped with ISVs in phylogenetic analyses. Some of these viruses were isolated in cell culture and we found no indication for the ability to infect vertebrate hosts. All of the isolated viruses were temperature-sensitive and did not replicate in the tested vertebrate cell lines including the immunodeficient Vero cell line. Only four of the detected 49 viruses may also be able to infect vertebrates according to placement in phylogenetic analyses (the two peribunyaviruses Cimo peribunyavirus I and II, the phenuivirus Cimo phenuivirus V and the orbivirus Wanken orbivirus). However, these four viruses were rarely detected (in total between one and five strains were found) and not included in the abundance analyses. The abundance analyses were based on viruses that were found at least ten times in the sample set. This was fulfilled for nine viruses. Five of these nine viruses could be isolated in cell culture and infection experiments as well as phylogenetic analyses indicated an insect-specific transmission cycle. The four viruses, which could not be isolated in cell culture, are closely related to previously described insect-specific viruses. We are therefore convinced that the data presented are robust and not influenced by vertebrate hosts.

8. You report an anthropogenic disturbance gradient from primary forest to village habitat but how was this quantified? How is a village more disturbed than an agricultural field (rice plantation?)? The method to rank these study sites, which becomes important for the analysis, was not described in the methods.

Our anthropogenic disturbance gradient was defined by apparent presence and density of humans as given by housing, e.g. camps and villages. In addition, we accounted for the degree of plant species turnover from primary forest to agriculture. The camp is an exception, as it is surrounded by primary forest and thus has a similar vegetation as the primary rainforest; however, we rated human presence higher and therefore grouped the camp closer to the village (Figure 1 a).

We added this explanation to the methods section (lines 508 – 512):

“The anthropogenic disturbance gradient was defined by apparent presence and density of humans as given by housing, e.g. camps and villages. In addition, we accounted for the degree of plant species turnover from primary forest to agriculture. The camp is an exception, as the vegetation is similar to the primary forest; however, we rated human presence higher and therefore grouped the camp closer to the village (Figure 1 a).”

9. Also, along this topic of study sites, it appears you really only had one replicate of each of the study site type. To test these hypotheses on how host communities influence viral communities it would seem prudent to have had multiple replicates of each study area.

The sampling design was balanced with different numbers of replicates per study site defined by the number of trap nights, so that the total sampling effort per site was approximately the same. The different replicates were then pooled per site for the analysis. However, we did not treat the replicates in our habitats separately in the model, but the 430 pools for several reasons. We did so, because the replicates themselves were not blocks of repetitions, but contained variance due to different trap heights in the canopy in some replicates, and e.g. for agriculture different replicates in plantations and arable fields. However, we wanted to elucidate the overall effect of human influence, and therefore we pooled the replicates per habitat, with the single pools with distinct mosquito species treated as replicates in the analyses. The field study with the different sampling locations is described in detail in Junglen et al. 2009 EcoHealth (49).

10. A suggested important contribution of the paper is the finding of an "abundance effect", suggesting that higher prevalence in degraded ecosystems is the result of host abundance, but additional ecological information is missing on the potential mechanisms leading to this effect. Breeding sites may be a main source of variation in species composition and abundances among habitats, but no comments on this are found on the manuscript.

Thank you for pointing this out. We added the following text to the discussion (lines 409 – 418):

“The ecological mechanisms leading to changes in mosquito community composition due to anthropogenic disturbance are presently poorly understood. Environmental changes, including land-use changes, but also changes in the host species composition have been suggested to act as drivers of vector-borne diseases, including important arboviral diseases like dengue (84-86). Yet, very few studies have focused on Culex nebulosus. It has been reported as a sylvatic species in Nigeria, Benin and Mozambique (87-89) but to the best of our knowledge this is the first study showing an increasing abundance of Culex nebulosus in modified ecosystems. Possible reasons for the varying mosquito community compositions in the different habitat types might be the availability of suitable breeding sites and food sources, the abundance of predators as well as differences in temperature and humidity (90-92).”

11. Some additional useful information could be provided to better understand mosquito sampling, for instance: the number of traps used, duration of sampling in each locality, and sampling dates to understand if there could be seasonal variation.

We now added the following section (lines 500 – 508):

“In total, 4562 female mosquitoes were collected in five habitat types along an anthropogenic disturbance gradient in the area of the Taï National Park in Côte d’Ivoire in 2004 (49). Habitat types represented pristine forest (PF), secondary forests (SF), agricultural areas (A), camps within primary forest (C) and villages (V) in the order of low to high human disturbance. Mosquitoes were collected using five CDC miniature light traps and one CDC gravid trap (developed by the U.S. Centers for Disease Control, John W. Hock Company, USA) per sampling site which were five sites in the PF, three sites in the SF, four sites in A (coffee, cacao, rice and rice plantations), four camp sites (C) and two villages (V). Sampling was conducted for three consecutive days per sampling site from February until June 2004.”

12. Mesoniviruses: Some mesonivirus species have been shown to be expectorated by mosquitoes, which explains their presence in multiple mosquito genera and suggests multiple modes of transmission. Perhaps insect-specific bunyaviruses could be the same given their detection in multiple mosquito genera?

This is a plausible hypothesis and should be studied in the future. Many thanks for mentioning this. Since this mechanism has been reported for some mesoni- and flaviviruses, as well as unclassified bunyaviruses, we added the following part to the respective part in the introduction (lines 99 – 100, also see response 1):

“It has been shown that at least some flavi-, mesoni and bunyaviruses are expectorated during feeding which may provide a route for horizontal virus transmission (44-47).”

13. Please clarify whether the numbers of pools of each mosquito species collected at each site are the same as in Table 1 of ref 29. I wanted to be able to see the actual numbers of mosquitoes collected at each location.

We confirm that the actual numbers of mosquitoes collected at each location is given in Table 1 of the former ref. 29, now ref 49. This table includes all collected mosquito specimens. However, not all collected mosquitoes were available for virus testing and used for other purposes, e.g. used as reference material. The total number of mosquitoes tested for infection with viruses per sampling location is provided in the methods section of this manuscript.

14. In line 97, it is stated that the aim was to capture all major taxa of arthropod-associated viruses, but I note that no assays were included to detect negeviruses, which are a growing taxon of insect-specific viruses.

We are aware that negeviruses are an import taxon of insect-specific viruses that are frequently isolated from mosquitoes. We tried to establish a generic RT-PCR assay to screen the mosquito homogenates for infection with negeviruses but unfortunately without success. As performed for the other screening assays, we based our assays on viruses that have been isolated from this data set and all available publicly sequences. While negeviruses grow to high titres in cell culture and could easily be detected by our assay in cell culture supernatant, it was not possible to amplify viral genome fragments from the mosquito homogenates. The amount of genome copies in primary mosquito material seems to be too low to allow for virus detection in primary material using generic RT-PCR. Therefore, it was not possible to include negevirus for the analyses of this study. We now added the following sentences to the manuscript (lines 528 – 529):

“Due to difficulties in the establishment of a generic RT-PCR assay for the unclassified taxon of negeviruses, it was not possible to include these viruses in the analyses.”

15. The Results section is too long, particularly with detailed virologic information, and includes parts that should go on the discussion.I strongly suggest to include SI Figure 1 (Biodiversity analyses) as a figure in the text since the main conclusions are based on variation of mosquito communities´ composition, and this is the result showing it.

We now moved Figure SI 1b to the main section of the manuscript and moved the rarefaction figure to the SI instead, as the latter could be described in the text. We agree that the result section is long but since this is an interdisciplinary journal the readers may find it helpful to receive some background information on the detected viruses. We thus decided not to remove this information from the results.

16. Ln. 128: The authors state "primer and sequences and cycling conditions are available upon request". If this is the first study to present the results from these newly designed primers, they should be included in this publication. Perhaps as supplemental material. In fact right not it is hard for me to know if you had 8 primer pairs for these RT-PCRs or if any targets overlapped and shared a forward sequence for example. In the following sentences you say you used previously published assays as well, which appear to be a combination of conventional and real-time assays, so now it is getting hard to untangle which results came from which PCR.

This is true and we apologise for not being clear in the beginning. All generic screening assays were performed as conventional nested (full and semi nested) RT-PCRs. This information is now included in the manuscript (lines 521 – 528). In addition to the generic assays, three specific qPCRs were performed to distinguish simultaneous infections with different viruses that could be targeted by the same conventional RT-PCR. The primers of the unpublished assays are now listed in Supplementary file 3.

Reviewer #1 (Recommendations for the authors):The authors have done an excellent job of displaying very complex data, but I do believe that the methods for the transmission of these viruses in nature needs to be taken into consideration. Granted though, that there are likely to be differing scenarios for different viral families. E.g. Flaviviruses: Numerous groups have shown vertical transmission as the mechanism for persistence of insect-specific flaviviruses in nature. Furthermore most insect-specific flavivirus species have been detected in only one species of mosquito suggesting strong evolution of the virus with their mosquito host.Mesoniviruses: Some mesonivirus species have been shown to be expectorated by mosquitoes, which explains their presence in multiple mosquito genera and suggests multiple modes of transmission. Perhaps insect-specific bunyaviruses could be the same given their detection in multiple mosquito genera?

This has been answered above. Please refer to point 12 for mesoni- and bunyaviruses. Regarding the ecology and transmission of ISVs please refer to points 1-4.

Please clarify whether the numbers of pools of each mosquito species collected at each site are the same as in Table 1 of ref 29. I wanted to be able to see the actual numbers of mosquitoes collected at each location.

This has been answered above. Please refer to point 9.

In line 97, it is stated that the aim was to capture all major taxa of arthropod-associated viruses, but I note that no assays were included to detect negeviruses, which are a growing taxon of insect-specific viruses.

This has been answered above. Please refer to point 14.

For many of the new viruses presented in this study, that have not been published previously, relatively small sequences of 500 – 1200 nt have been used to assign viruses as new species. In terms of the flaviviruses, it is generally considered that a new virus species must have >16% sequence identity (nt) to be considered a distinct species.

This has been answered above. Please refer to point 6.

Please clarify that in Figure 8, that the left hand graph is all detections of the virus, despite the mosquito species – this didn't come across clearly in the figure legend.

Thank you for this valuable comment. We changed the first part of the figure legend accordingly:

“For all viruses, that were detected in >10 pools, GOLV (A), HEBV (B), FERV (C), Cimo rhabdovirus I (D), CAVV (E), Cimo phenuivirus II (F), Jonchet virus (G), Spilikins virus (H) and Cimo flavivirus I (I), the MIR and MLE per 1000 mosquitoes of the whole data set was calculated for all habitat types (left graphs for A-E).”

Were there any attempts to obtain isolates of Cimo peribunyaviruses I and II?

We tried to isolate these two novel viruses in different insect and vertebrate cell lines but no replication was observed in any of the cell lines by testing the cell culture supernatant of four passages by PCR (see Methods “Viruses were isolated in cell culture as described before (49).”). We added information on negative isolation attempts in cell culture to the text (line 340):

“All other viruses could not be isolated in cell culture.”

Reviewer #2 (Recommendations for the authors):The Results section is too long, particularly with detailed virologic information, and includes parts that should go on the discussion.I strongly suggest to include SI Figure 1 (Biodiversity analyses) as a figure in the text since the main conclusions are based on variation of mosquito communities´ composition, and this is the result showing it.

This has been answered above. Please refer to point 15.

Reviewer #3 (Recommendations for the authors):Ln. 21: Consider change to "…on the mechanism of emergence."

The sentence was changed to:

“Previously unknown pathogens often emerge from primary ecosystems, but there is little knowledge on the mechanisms of emergence.”

Ln. 22: Consider change to "...often observe contradictory results."

Done.

Ln. 27: "was" should be "were".

Unchanged, as the “was” refers to the “majority” and not to “viruses”: “The majority of the 49 viruses was detected with low prevalence.”

Ln. 28: You reference congruence to the dilution effect hypothesis here. But then the final statements of the discussion say 'No general dilution or amplification effect was observed'. Given the comments above, I have trouble seeing evidence to support conclusions on this topic.

We observed an increase in prevalence in disturbed habitats for several viruses which would be congruent with the dilution effect. But likewise, we observed viruses with a high prevalence in pristine or intermediately disturbed habitats that rarely occurred in disturbed habitats. Therefore, we observed no general dilution or amplification effect but a mixture of different prevalence patterns.

Ln. 128: The authors state "primer and sequences and cycling conditions are available upon request". If this is the first study to present the results from these newly designed primers, they should be included in this publication. Perhaps as supplemental material. In fact right not it is hard for me to know if you had 8 primer pairs for these RT-PCRs or if any targets overlapped and shared a forward sequence for example. In the following sentences you say you used previously published assays as well, which appear to be a combination of conventional and real-time assays, so now it is getting hard to untangle which results came from which PCR.Table 1. Would be good if the table legend or footnotes could explain the abbreviations in the column headers. I think these refer to study sites. Is there a way to sum these study sites and present a total number of unique viruses from each of the study sites? Would this be the number displayed in Figure 7A?

We included the following sentence in the legend of table 1 to explain the abbreviations:

“Abbreviations are PF, primary forest; SF, secondary forest; A, agriculture; C, camp and V, village.”

The total number of unique viruses from each study site is displayed in Figure 7A.

Ln. 260. You are using the term 'insect-restricted' here and in a couple other locations. Is this the same as 'insect-specific'? If so, I suggest staying consistent with insect-specific which you already use elsewhere in the manuscript. Or you could say these viruses are restricted to mosquitoes if that is appropriate in context.

Thank you for this valuable comment. We changed the term “insect-restricted” to “insect-specific” in the whole manuscript.

Figure 2. For all these phylogenetic trees, it is hard to see how large of a gene region (or entire gene?) was used to build these. Would be helpful to include the number of base pairs, either in legend or in methods section.

The respective alignment size was included in all figure legends of phylogenetic trees.

Ln. 356. I'm not familiar with this process to attempt to identify if a virus is insect-specific by increasing temperature using insect cells (C6/36). But this sounds strange because holding C6/36 at 34C doesn't sound good for the cells so did the virus not replicate because the cells were all dead? Do you have a control of some kind showing C6/36 are still healthy at 34C? Did you attempt to grow these viruses in vertebrate cells?

This method was described by Marklewitz et al. (PNAS; 2015; doi: 10.1073/pnas.1502036112). The C6/36 cells remain viable up to 34°C and arboviruses such as Rift Valley fever virus and La Crosse virus can still replicate in C6/36 cells at 34°C. We included this reference:

“The cells were infected at a MOI of 0.1 and incubated for 3 days at 28, 30, 32 and 34°C to determine temperature sensitivity as previously described for dual-host and insect-specific bunyaviruses (36).”

We also tried to infect vertebrate cells with these viruses but they did not replicate in any of the cell lines tested.

Ln. 365. You are presenting the detection of viral integration into the mosquito genome but it is not clear how these observations were treated in the context of the analysis of viral taxonomy across the study sites. I assume there were ignored.

The potentially integrated virus-like sequences were not included in the diversity and prevalence analyses. They are only shown in Table 1 and characterized in the section “Detection of non-retroviral integrated RNA virus sequences.” To clarify this, we included the following sentence:

“These findings were not included in the virus diversity and prevalence analyses.”

Ln. 394. Is it better to say "…pools were found positive at least one virus"?

Done.

Figure 7: Once again, would be good to identify what the habitat abbreviations refer to in legend.

Done.

Ln. 560. Chikungunya should be lower cased.

Done.

Ln. 609-611. True, so you should pull in the relevant literature from plant viruses (and in particular grasses) which have been used as a system to study the relationship between viral prevalence and host community composition.

Thank you. These references have been added including the following text:

“For example, (i) a low diversity of grassland plant species diversity was associated with a higher abundance of fungal plant pathogens (110), (ii) the disease and infection risk of populations of wild pepper (chiltepin) with different viruses increased with the level of human management, which was associated with decreased species diversity and host genetic diversity, and increased host plant density (111), and (iii) the prevalence of four generalist aphid-vectored pathogens (barley and cereal yellow dwarf viruses) increased in grass host plants as local host richness declined indicating that virus incidence in the focal host species is affected by the composition of the entire host community rather than by dependence on species diversity (110-112).”